# PROCEEDINGS A

civil engineering, environmental engineering

direct shear, slope stability, root reinforcement, digital volume correlation, X-ray computed tomography, shear zone thickness

**Author for correspondence:**
Joel A. Smethurst
e-mail: jas@soton.ac.uk

# Modelling of stress transfer in root-reinforced soils informed by four-dimensional X-ray computed tomography and digital volume correlation data

Daniel J. Bull[1], Joel A. Smethurst[1], Gerrit J. Meijer[2], I. Sinclair[1], Fabrice Pierron[1], Tiina Roose[1], William Powrie[1] and A. Glyn Bengough[3,4]

[1]School of Engineering, Faculty Engineering and Physical Sciences, University of Southampton, Southampton SO17 1BJ, UK
[2]Department of Architecture and Civil Engineering, University of Bath, Bath BA2 7AY, UK
[3]School of Science and Engineering, University of Dundee, Dundee DD1 4HN, UK
[4]The James Hutton Institute, Dundee DD2 5DA, UK

DJB, 0000-0001-6711-6153; JAS, 0000-0001-8175-985X; TR, 0000-0001-8710-1063

Vegetation enhances soil shearing resistance through water uptake and root reinforcement. Analytical models for soils reinforced with roots rely on input parameters that are difficult to measure, leading to widely varying predictions of behaviour. The opaque heterogeneous nature of rooted soils results in complex soil–root interaction mechanisms that cannot easily be quantified. The authors measured, for the first time, the shear resistance and deformations of fallow, willow-rooted and gorse-rooted soils during direct shear using X-ray computed tomography and digital volume correlation. Both species caused an increase in shear zone thickness, both initially and as shear progressed. Shear zone thickness peaked at up to 35 mm, often close to the thickest roots and towards the centre of the column. Root extension during shear was 10–30% less than the tri-linear root profile assumed in a Waldron-type model, owing to root curvature. Root analogues used to explore

the root–soil interface behaviour suggested that root lateral branches play an important role in anchoring the roots. The Waldron-type model was modified to incorporate non-uniform shear zone thickness and growth, and accurately predicted the observed, up to sevenfold, increase in shear resistance of root-reinforced soil.

## 1. Introduction

Vegetation can offer a low-cost solution for improving soil stability on slopes through root reinforcement mechanisms [1–5]. Many laboratory experiments and field studies have demonstrated increases in the effective shear resistance of soils associated with root reinforcement [6–14]. However, little is known about the exact mechanisms of soil–root reinforcement. Part of the complication arises from the large number of parameters involved, which include: size and distribution of roots, variability in root strength and stiffness, soil conditions (compaction, stress levels, moisture content, etc.) and root failure mechanisms (breakage, slippage and buckling) [15,16]. This is further compounded by the difficulty in observing such behaviour within the bulk of the soil. While there have been numerous experimental efforts to observe these behaviours at the boundaries, e.g. through a transparent window [17], many of the mechanisms are controlled by behaviour occurring within the bulk of the soil, which is difficult to capture using traditional surface-based observation methods.

To obtain a better understanding of mechanisms within the soil, there has been an increase in the use of X-ray computed tomography (XCT) by the soil science and geotechnical communities over the past decade. This technique offers a non-destructive approach to observing geometric features and mechanisms that are hidden within the bulk of the soil. When coupled with time-series experiments and digital volume correlation (DVC) techniques, it is possible to obtain a direct measure of local three-dimensional displacements and strains within the volume [18–20]. DVC works by dividing the three-dimensional XCT scan volume into smaller subsets and tracking the local positions of these subsets during loading to obtain a displacement vector map. The subsets require sufficient patterns of data (e.g. grains of soil) to calculate their local positions. From the displacement field, it is possible to derive a strain field. The feasibility of experiments applying DVC techniques to direct shear tests has been demonstrated in a previous study [21].

Various analytical modelling approaches to quantify root reinforcement behaviour in soils in direct shear have been proposed. One of the most well-known is referred to as the Waldron and Wu model [22,23], which considers the tensile force generated in an elastic root during its deformation within a shear zone to determine the shearing resistance of the rooted soil. Since the introduction of the Waldron–Wu approach, there have been numerous developments and adaptions including fibre-bundle models capturing progressive root failure [15,24–30].

Part of the challenge with these analytical models is their reliance on accurate input parameters from experiments. Certain parameters are relatively straightforward to obtain. For example, root diameters and root area fractions can be measured from roots that cross the shear plane surface and root stiffness obtained from uniaxial tensile testing [9]. However, other parameters, such as shear zone thickness and soil–root interface shear stress, are much more difficult to determine [17].

Previous attempts at measuring shear zone thickness or shear banding behaviour have involved observing behaviour at the boundary through a transparent window using root analogues [17], image processing strategies [31] and digital image correlation approaches [32–34]. The shape of the shear deformation pattern can make it difficult to establish an unambiguous measure of shear zone thickness. Human interpretation when taking measurements can also lead to inconsistent measurements [17]. Additionally, little is known about shear zone thickness behaviour within the bulk of the soil and local effects near the presence of roots. Because of these unknowns, coupled with the difficulty in measuring shear zone thickness, many analytical models use empirical approaches to estimate the shear zone thickness and make assumptions about the shear zone, e.g. that it remains constant in size.

The limiting soil–root interface stress ($\tau'$ in the Waldron model) can be taken as purely frictional, based on the normal stress within the soil and an estimate of the interface friction angle. However, this neglects root branching and lateral roots that anchor the root system into the soil and are likely to increase $\tau'$ [21,22]. Appropriate values of $\tau'$ that take into account these effects will vary with plant root morphology and soil type, and are not well characterized.

In this paper, three-dimensional measurements of local displacements and strains during direct shear tests conducted on root-reinforced soils (willow and gorse) and unrooted (fallow) test specimens are presented. By conducting direct shear tests within an XCT scanner and applying DVC, the chronology of geometrical measurements within the bulk of the soil is captured. To complement the XCT work, detailed conventional measurements of root position and size at the shear plane are made. Root analogues are studied to understand the effect of root friction and root anchorage on soil reinforcement. The experimental aspects of this research are assessed in the context of the Waldron analytical model to understand the influence of certain mechanisms on root reinforcement performance and assess the model's predictive capability for root-reinforced soils.

## 2. Methodology

### (a) Specimen preparation

Soil specimens were prepared within a 110 mm outer diameter (103 mm inner diameter), 500 mm long cylindrical tube. The tube comprised two 250 mm long sections supported in the middle by a temporary bracket to leave a 2 mm clearance gap. Bullionfield soil (71% sand, 19% silt and 10% clay, with a pH of 6.2) [9] was supplied by the James Hutton Institute, Dundee, UK. The soil was oven dried at 60°C for 48 h, pulverized and sieved to remove grains greater than 2 mm. De-aired water was added to achieve a water content of 0.18 g g$^{-1}$, and the soil was thoroughly mixed before being stored in a sealed plastic box overnight for the moisture to equalize through the soil. The soil was compacted in 10 equal layers, 50 mm in height, using a tamper of a diameter slightly smaller than the tube, to give a bulk dry density of 1.4 Mg m$^{-3}$. The soil surface was scarified before placement and compaction of the next layer to avoid any obvious structure in the specimen from the compaction process.

Two plant species were chosen: willow (*Salix viminalis*, variety Tora) and gorse (*Ulex europaeus* L.). A single plant was grown in each specimen tube for 60 (willow) and 120 days (gorse) under artificial lighting comprising a Maxibright T5 unit, equipped with eight blue T5 fluorescent tubes delivering 4450 lumens of light per tube, operating for 16 h per day. The height of the lights was adjusted to maintain approximately 150 mm distance between the lights and tallest part of the plants.

A total of five replicates of willow, gorse and unrooted (fallow) specimens were prepared. Two were tested in direct shear while being XCT scanned, and three were tested in direct shear outside the XCT scanner (table 1: the original specimen names have been retained to maintain traceability to the raw data). Available resources meant that it was only possible to XCT scan two replicates, with the other three replicates providing additional information about the variability in root area and shearing behaviour. Prior to testing, the plant stem was cut off at the soil surface to eliminate transpiration (except for Willow F, where it was left on), and specimens were saturated in a water bath before being placed on a bed of saturated sand within a large container. Water was drained from the bottom of the container to a water table in the sand 0.5 m below the shear plane in the specimen tubes, resulting in a matric water suction of about 5 kPa at the shear plane.

To investigate the root–soil interface shearing resistance and the potential importance of root branching in anchoring the roots within the soil, three further specimens were created containing root analogues. The first specimen (Analogue M) contained a centrally located vertical artificial smooth root fully extending the full 500 mm depth of the tube. The second (Analogue Q) contained a vertical artificial root 200 mm long, constrained by two 50 mm diameter, 2 mm thick, aluminium disc anchors placed at each end, 100 mm above and below the shear plane. The discs

**Table 1.** Specimen names and test conditions.

| tested in XCT scanner (interrupted shearing) | tested outside the XCT scanner (continuous shearing) |
|---|---|
| Willow C | Willow E |
| Willow F | Willow H |
| Fallow D | Willow I |
| Fallow P | Gorse B |
| Gorse A | Gorse J |
| Gorse G | Gorse K |
| | Fallow L |
| | Fallow N |
| | Fallow O |
| | Analogue M |
| | Analogue Q |
| | Analogue R |

were intended to represent side branches acting as anchors, holding the root in position. The third specimen (Analogue R) contained two plastic filaments, again extending between two aluminium discs. The analogue roots were made from a cylindrical ABS plastic filament of 1.75 mm diameter. The root analogues were installed as the soil was compacted into the specimen tube using a tamper with a hole cut in its centre. In all other respects, the specimens were prepared in the same way as described above. The analogue specimens were tested in direct shear outside the XCT scanner.

## (b) *In situ* XCT experiments

The prepared specimen tubes were mounted inside a bespoke direct shear test rig (figure 1; [21]) designed to be operated within a large, walk-in XCT scanner. The rig allowed XCT scans to be taken after each of a series of incremental shear displacement steps, while sensor data were recorded simultaneously [21].

The direct shear test rig fully constrained the upper portion of the soil tube and allowed a single degree of freedom (displacement along the $x$-axis) in the lower soil tube. In addition to the shear stress and the applied displacement, soil pore water matric suctions were measured using a tensiometer (Delta-T Devices model SWT5) with the point of measurement 50 mm below the soil surface. The tensiometer measurements were to check that any significant plant-generated suctions were removed in the specimen saturation process, that the suctions were consistent between specimens and remained stable during the up to 12 h long XCT experiments. Figure 1 shows the right-handed coordinate system and origin (located at the centre of the tube) used as a reference point for all measurements presented in this paper.

The upper and lower portions of the tube were mounted on the top and bottom bracket within the test rig, respectively. The temporary bracket supporting both halves of the tube was then removed. A razor blade, set to a depth of 2 mm, was run round the perimeter of the tube at the shear plane to sever and remove any influence of roots in direct contact with the sidewall of the tube. To minimize the loss of water during the experiment, the top and bottom of the tube were covered and taped with aluminium foil to provide an airtight seal. No additional normal stress was applied to the soil column, so that the normal stress at the shear plane was due only to the self-weight of the soil.

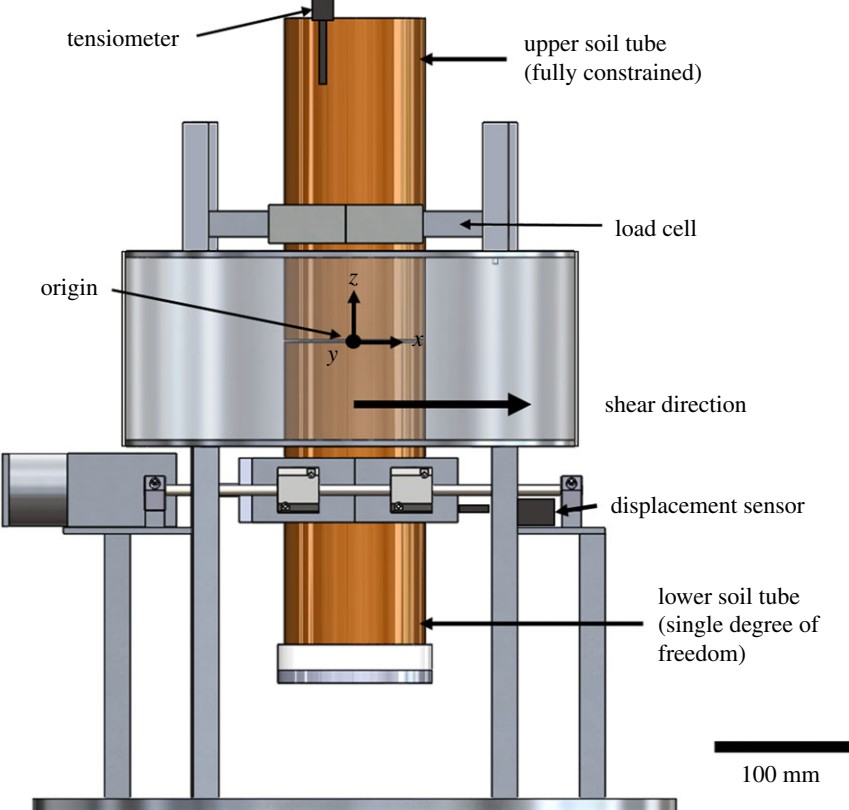

**Figure 1.** Schematic diagram of direct shear test rig. A photograph of the rig within the XCT scanner is given in [21]. (Online version in colour.)

Table 2 details the horizontal shear displacements applied at each displacement step. The XCT scan parameters used are given in table 3. These were selected carefully to achieve a balance between the XCT field-of-view, voxel resolution, image quality (contrast-to noise ratio), DVC process and scan time [21]. The XCT field of view was approximately $80 \times 80 \times 80$ mm centred on the origin (figure 1) and did not extend to the full width of the specimen (of 103 mm). Test specimens were first XCT scanned undeformed, followed by a scan after each of seven applied displacement steps with a target increase in shear displacement of approximately 3 mm per step. The initial applied steps were not of a consistent size as it was difficult to set up the specimen in the apparatus without some slack in the system, and displacement was later zeroed at the point at which load was first measured. Later applied steps were all 3.3 mm, until the maximum travel was reached on the shear apparatus at around 20 mm. The shear displacement loading rate was 1 mm minute$^{-1}$. A 30 min wait period following each displacement step allowed for any stress/strain relaxation to occur within the rooted soil prior to XCT scanning, to minimize movement artefacts during the XCT scans. Further information on the apparatus and experimental procedure is given in [21].

## (c) DVC

DVC was carried out on the XCT scan volumes using DaVis LaVision v. 8 software [35]. Table 4 details the DVC parameters used. DVC processing involved correlating sub-volumes from within a deformed volume against a reference volume to calculate local displacements and subsequently derive strain information. After DVC processing, full-field data containing displacements, normal

**Table 2.** Details of applied *x*-displacement (mm) at each displacement step during *in situ* XCT test for each specimen.

| specimen | horizontal *x*-displacement (mm) | | | | | | |
|---|---|---|---|---|---|---|---|
| | step 1 | step 2 | step 3 | step 4 | step 5 | step 6 | step 7 |
| Willow C | 1.72 | 4.87 | 8.15 | 11.39 | 14.69 | 17.93 | 19.83 |
| Willow F | 1.23 | 4.20 | 7.44 | 10.65 | 13.82 | 17.03 | 18.75 |
| Fallow D | 2.17 | 5.48 | 8.80 | 12.14 | 15.47 | 18.76 | 20.84 |
| Fallow P | 3.17 | 6.47 | 9.79 | 13.11 | 16.41 | 19.73 | 20.46 |
| Gorse A | 1.49 | 4.73 | 8.03 | 11.34 | 14.63 | 17.95 | 20.14 |
| Gorse G | 0.77 | 4.00 | 7.31 | 10.59 | 13.89 | 17.19 | 19.31 |

**Table 3.** XCT scan parameters for *in situ* direct shear testing.

| XCT scan parameter | value |
|---|---|
| energy (peak) | 300 kVp |
| power | 90 W |
| voxel resolution | 46 μm |
| field of view | 80 × 80 × 80 mm |
| number of projections | 3142 |
| source to object distance | 359 mm |
| source to detector distance | 1546 mm |
| frames per projection | 4 |
| exposure time | 134 ms |
| scan time | 30 min (per applied displacement step) |
| filtering | 6.6 mm aluminium (total transmission through wall of cylindrical support on test rig) |

strain and shear strain components were post-processed using a custom MATLAB script. A study was carried out to optimize the DVC sub-volume size for reliable correlation, error caused by noise, strain measurement accuracy and spatial resolution [21]. This found that it was necessary to correlate adjacent scans in the XCT dataset (rather than e.g. always correlating back to the first scan), with the implication that the noise from each correlation step would sum through the test. The minimum subset size achievable was 32 pixels cubed, and for subsets of this size noise was found to accumulate linearly with the number of displacement steps in the sequence, with the largest strain standard deviation reaching 38 millistrain at the end of the test. The subset size and noise sensitivity study is described in detail in [21]. The 75% overlap in the subset size meant that displacement was calculated on a three-dimensional grid of eight pixels or 0.37 mm.

The DVC-calculated displacements were also used to estimate the size of the shear zone within the sample, as this is an important input parameter for many analytical root models. The large amount of data generated by DVC (1.6 million data values per displacement step), and the need to provide a consistent method of determining shear zone thickness, led to the development of an automatic process for calculating local shear zone thickness. To determine the local shear zone thickness at each (*x*,*y*) position on the shear plane, a least-squares regression fit was used to generate a tri-linear profile to *z*-position versus *x*-displacement data taken at each (*x*,*y*) position, with the fit above and below the shear zone always a vertical line. The shear zone thickness was calculated by taking the *z*-distance between the two knee points on the tri-linear profile (figure 2*b*). This method was chosen after four different calculation approaches were trialled

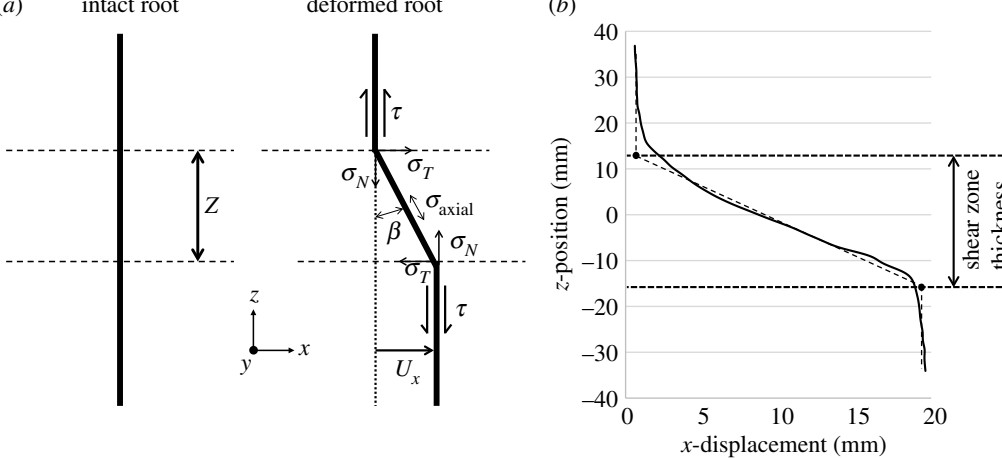

**Figure 2.** (*a*) Schematic illustrating the development of root stresses during direct shear [22]. (*b*) Example of tri-linear fit (dotted line) used to determine local shear zone thickness from *z*-position versus *x*-displacement (solid line) DVC data. The shear zone thickness was calculated at every (*x,y*) position on the shear plane, for each displacement step in the test.

**Table 4.** DVC analysis parameters [36].

| | |
|---|---|
| DVC software | DaVis LaVision, Version 8 |
| image filtering | Gaussian filter with a $3 \times 3 \times 3$ pixel kernel |
| subset sizes | $32^3$ pixels$^3$ |
| | $1.47^3$ mm$^3$ |
| subset step size | 75% overlap |
| subset shape function | affine |
| matching criterion | cross-correlation (CC) |
| interpolant | Spline 6 |
| strain calculation method | centred finite difference |
| strain window | 3 data points |
| virtual strain gauge size | 48 pixels |
| | 2.2 mm |
| strain formulation | Green-Lagrange |
| noise for $32^3$ pixels$^3$ subset size | displacement noise: 5.4 µm |
| | Strain noise: 6.8 millistrain |

(which are described in the electronic supplementary material). As with the other approaches tested, the method may not identify quite the same knee points as visual inspection might, but it was found to produce a very consistent measurement of shear band height within the (*x,y*) space that allowed the size variation across the sample (from 2 to 30 mm in height) and the general growth trend to be clearly identified.

## (d) Root segmentation and root count on the shear plane

For the sample size and XCT set-up (table 3) used, the resulting XCT scans were found to capture the soil grains and roots larger than 1.5 mm in diameter. Owing to the voxel intensity variations of the roots and their similarity to the voxel greyscales in the surrounding soil, it was found to be

difficult to distinguish many roots smaller than about 1.5 mm/30 voxels in diameter. Automated root segmentation methods trialled on larger roots were unsuccessful, requiring the larger roots to be manually segmented from the volume. An example of a detailed XCT image from a specimen and further comment on the root segmentation is given in [21]. As a result of the inability to fully segment the root system within the XCT volume, following each direct shear experiment the shear plane was examined using a macroscope, and the location and diameter of roots crossing the shear plane was measured. This enabled a reliable measure of the full root area ratio (RAR) (i.e. the total root cross-sectional area per total unit area of soil [the area of soil particles, voids and roots]) on the shear plane of each specimen to be obtained, which could then be related to the shearing resistance of the specimen.

## (e) Analysis of root and soil deformation behaviour

The results obtained from the XCT shear experiments, including the shearing resistance, deformed segmented roots, RAR and DVC results including the height of the shear zone, were used to explore the accuracy of the simple root model described by Waldron [22]. The Waldron approach is based on the hypothesis that root reinforcement of the soil, as illustrated in figure 2a, occurs through two components:

1. The root tensile stress in the direction of shearing ($\sigma_T$), which adds to the shear resistance directly.
2. The root tensile stress normal to the shear plane ($\sigma_N$), which generates additional normal stress on the soil shear plane, resulting in additional soil shear resistance through friction.

In equation form, the increase in soil shear resistance due to the addition of roots, $\Delta S$, is obtained by summing over all roots $i$:

$$\Delta S = \sum_i a_r k (\sec \beta - 1)^{0.5} (\sin \beta + \cos \beta \tan \phi'), \tag{2.1}$$

$$k = \left( \frac{4\tau' Z E}{D} \right)^{0.5} \tag{2.2}$$

and
$$\beta = \tan^{-1} \left( \frac{U_x}{Z} \right), \tag{2.3}$$

where $a_r$ is the RAR, $\phi'$ is the internal angle of friction of the soil, $\tau'$ is the limiting bond or interface friction stress between the root and soil, $Z$ is the shear zone thickness, $E$ is the stiffness (Young's modulus) of the root, $D$ is the diameter of the root and $U_x$ is the applied shear displacement.

## 3. Results

## (a) Direct shear test results and root measurements

### (i) Shear displacement–shear stress results

Shear stress, shear displacement and soil suction data were recorded for each of the specimens tested in direct shear. Figure 3 shows shear stress–displacement plots for willow (red), gorse (blue) and fallow (green) specimens, for tests carried out continuously and tests with interruptions for XCT scans. The traces show that some stress relaxation occurs during the interruptions in the shear displacement applied to the XCT-scanned specimens, although this appears to have no effect on the overall shearing behaviour of the specimen when compared to samples that are sheared continuously [21]. Table 5 shows the measured bulk dry soil density, soil pore water matric suction, the water content estimated from the measured suction and a previously determined soil water retention curve for the Bullionfield soil [9], and the RAR, for each test specimen. The bulk densities and suctions in the specimens are generally very consistent and

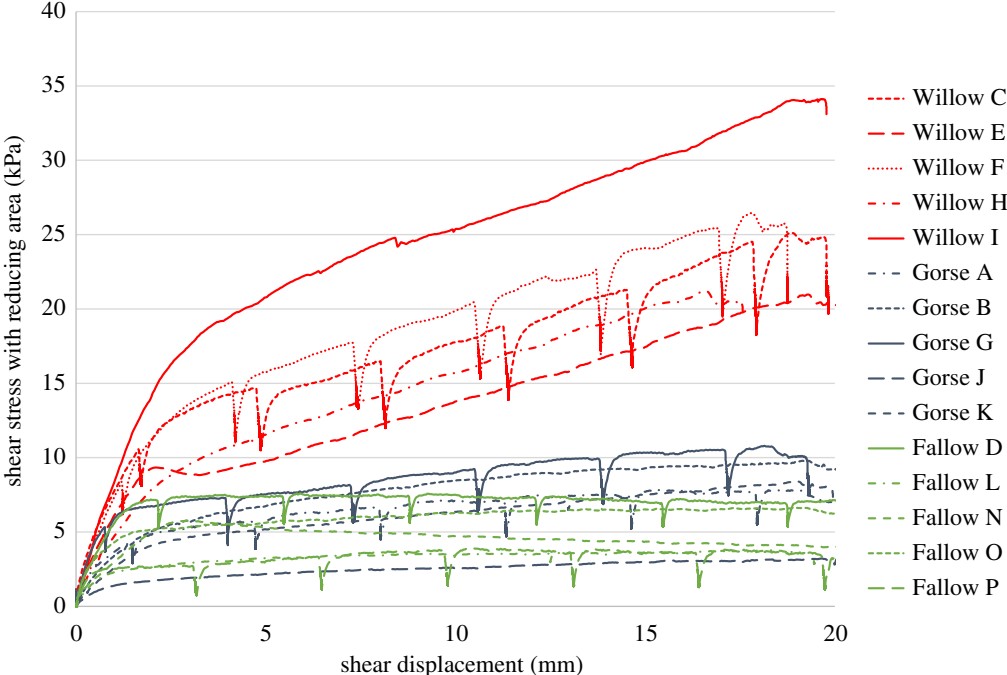

**Figure 3.** Direct shear stress–displacement plots for Willow (red), Gorse (blue) and Fallow (green) specimens. (Online version in colour.)

were controlled well by the experimental methodology; as may be expected there are differences in RAR between specimens caused by natural variability in plant growth [9].

Before making comparisons between individual tests, it is important to note variations between specimens that could have led to differences in shear stress versus *x*-displacement response. The Willow F specimen was tested with drier soil. During both the preparation for XCT scanning and the 12 h XCT experiment, the plant stem and leaves were left on the specimen. Transpiration led to this specimen drying and a soil suction greater than 88.2 kPa, beyond the measurement range of the tensiometer. This contrasts with the soil suction readings for all other specimens (between 2.0 and 3.6 kPa), where the plant stem was cut just above the soil surface prior to shear testing and the surface of the tube was covered. The effectiveness of this procedure in minimizing water loss was evidenced by the constant soil suction readings throughout the duration of these other tests. However, figure 3 shows that the greater soil suction and thus effective stress in the Willow F specimen does not appear to have a significant effect on the shear stress obtained. This may have been caused by differences in the pore water suction between the measurement location (the tensiometer was placed close to the top of the specimen; figure 1) and the shear plane.

The Willow I specimen exhibited a greater increase in shear stress compared with the other willow specimens. This is attributed to Willow I having approximately 40% more RAR than Willow F and Willow C specimens.

Gorse B and G specimens showed minor shear strength improvements over unrooted fallow specimens. Gorse A, J and K exhibited similar responses to unrooted fallow specimens, which is attributed to there being only shallow (less than 50 mm) root penetration beyond the shear plane, little to no root branching below the shear plane and very low RAR values (of less than 0.1%).

In the fallow tests, a scatter range of peak shear stress (3–7 kPa) was observed. This is despite compaction to a similar bulk dry density during soil preparation, and similar tensiometer readings (and implied water contents) during the direct shear tests (table 5). The XCT scans of Fallow D and Fallow P suggested that the larger shearing resistance from Fallow D was

**Table 5.** Soil dry bulk density, soil pore water matric suction and RAR measured for each test specimen, and corresponding soil water contents calculated using the matric suction and a water retention curve for the soil. RARs are measured from macroscopy taken on the shear plane post testing. A problem with the tensiometer sensor in the Analogue R test means that there are no suction data for this specimen.

| specimen | dry bulk density (Mg m$^{-3}$) | pore water matric suction (kPa) | water content (g/g) | root area ratio (%) |
|---|---|---|---|---|
| Willow C | 1.40 | 3.64 | 0.257 | 0.409 |
| Willow E | 1.41 | 2.57 | 0.261 | 0.275 |
| Willow F | 1.36 | >88.24 | <0.176 | 0.340 |
| Willow H | 1.43 | 2.81 | 0.260 | 0.272 |
| Willow I | 1.37 | 3.65 | 0.257 | 0.575 |
| Gorse A | 1.39 | 2.73 | 0.260 | 0.036 |
| Gorse B | 1.43 | 2.57 | 0.261 | 0.065 |
| Gorse G | 1.38 | 3.43 | 0.257 | 0.063 |
| Gorse J | 1.41 | 2.56 | 0.261 | 0.013 |
| Gorse K | 1.36 | 2.41 | 0.262 | 0.050 |
| Fallow D | 1.39 | 2.74 | 0.260 | — |
| Fallow L | 1.36 | 2.68 | 0.261 | — |
| Fallow N | 1.42 | 2.07 | 0.263 | — |
| Fallow O | 1.37 | 2.68 | 0.261 | — |
| Fallow P | 1.38 | 3.37 | 0.258 | — |
| Analogue M | 1.43 | 1.51 | 0.265 | 0.029 |
| Analogue Q | 1.41 | 1.71 | 0.264 | 0.029 |
| Analogue R | 1.42 | — | — | 0.058 |

possibly caused by 'peds' (agglomerations of individual grains held together by local suctions, unintentionally formed during specimen preparation) within the shear plane in the Fallow D specimen. This is considered further in the electronic supplementary material.

### (ii) Root count results

Figure 4 shows macrographs from three of the XCT-scanned specimens, and the plotted results of the root measurements from the shear plane of Willow C. Information about the diameter and location of each root on the shear plane was used to determine the RAR (table 5) and in later analysis of the shear tests using the Waldron model. In the willow specimens, there is a tendency for roots to be distributed towards the edge (within 20 mm of the tube sidewall). This is often associated with a pot-grown root system, where the roots may be deflected by the outer boundary. It also means that the RARs obtained are likely to be larger than for an equivalent plant grown without lateral restraint to the roots.

### (iii) Relationship between shear stress and root area ratio

Figure 5 shows the correlation between the peak shear stress achieved during 20 mm of relative shear with RAR. The positive linear trend agrees with the results of similar shear experiments [9,37] and the linear relationship between the additional shear resistance provided by the roots and the RAR given in the analytical approach in equation 2.1. The results show that the willow

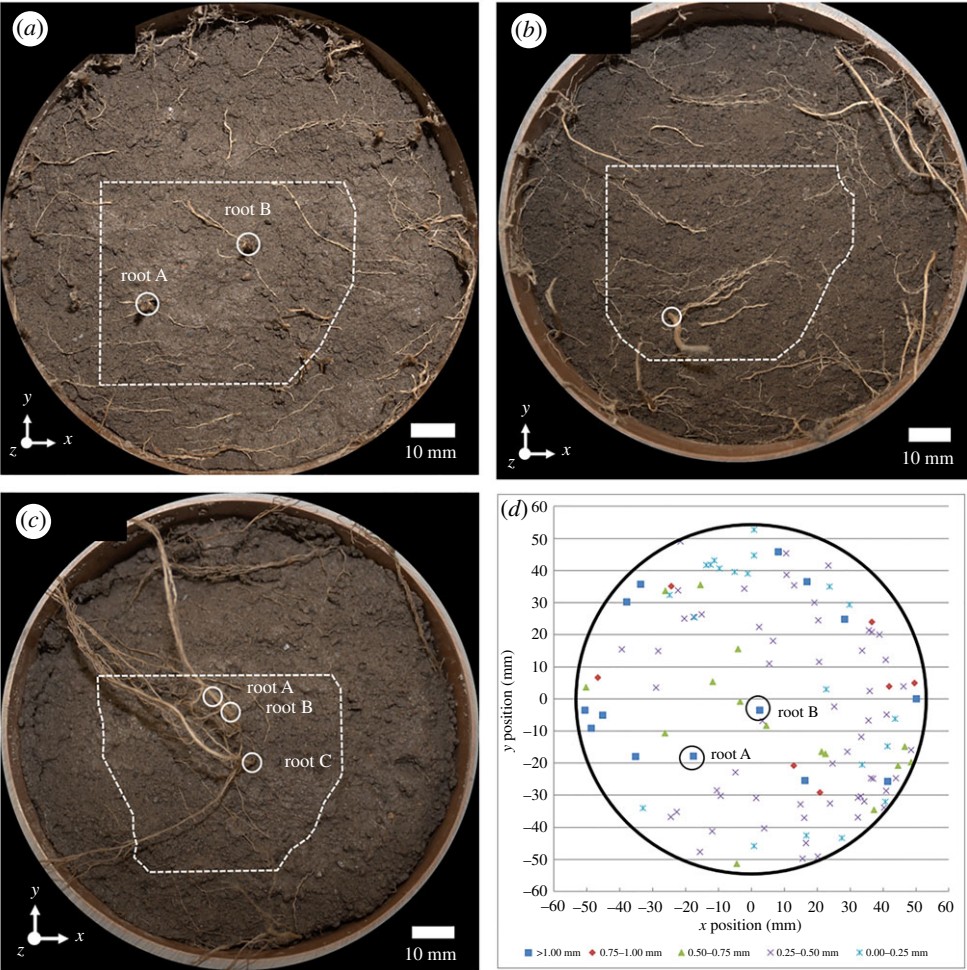

**Figure 4.** Macrographs of shear planes for rooted specimens. Gorse A was omitted as only fine shallow roots crossed the shear plane, which were pulled out during direct shearing. The dotted line overlay indicates the region boundary captured using DVC; circled regions are roots greater than 1 mm in diameter corresponding to local increases in shear zone thickness in figure 7. The roots imaged are those present in a 10 mm thick layer that was washed away in preparation for the macrograph; (*a*) Willow C; (*b*) Willow F; (*c*) Gorse G; (*d*) shows the recorded positions and diameters of the roots for Willow C. (Online version in colour.)

specimens gave the greatest shear resistance, primarily because they have RARs of an order of magnitude greater than the gorse tests.

Trend lines were fitted separately to willow and gorse, with $R^2$ values of 0.97 and 0.65, respectively. (Fallow data were included in both to account for behaviour without roots.) Owing to the low RARs and narrow range (0.01–0.07%), the gorse data were more sensitive than willow (0.27–0.57%) to scatter caused by slight variations in soil composition on the shear plane, as observed in fallow specimens; hence the lower $R^2$ value.

## (b) Shear zone thickness

### (i) Shear zone thickness: non-uniformity within the tube and near roots

During direct shear tests, a non-uniform shear zone thickness was observed. The shear zone formed a diamond shape when viewed side-on in two dimensionals, as shown in figure 6. This is consistent with other experimental [38] and numerical [39,40] observations of direct shear, in

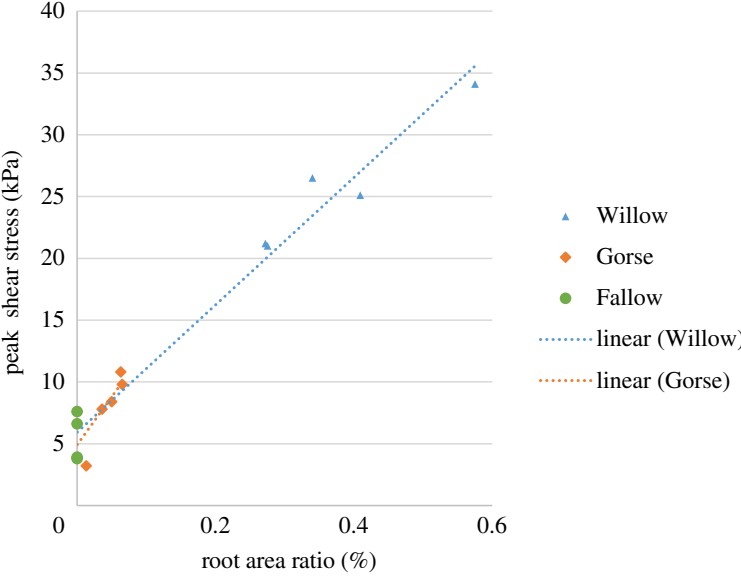

**Figure 5.** Effect of RAR on peak shear stress reached during direct shear tests. RAR was calculated by summing the cross-sectional areas of each root passing through the shear plane. (Online version in colour.)

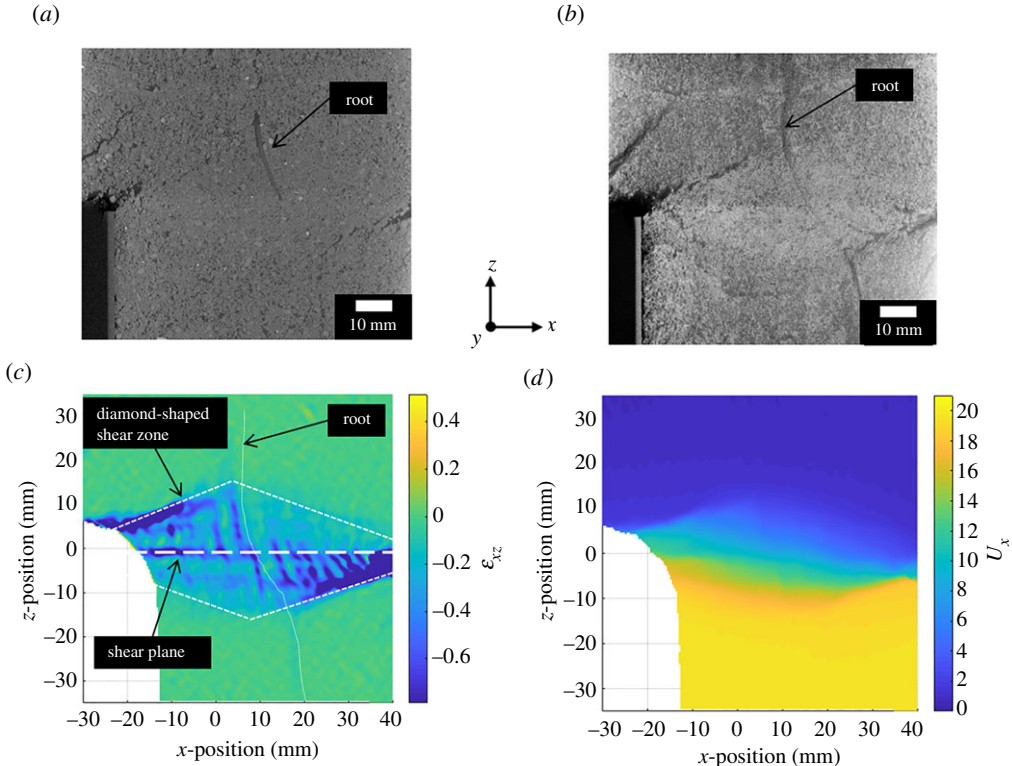

**Figure 6.** XCT cross-section (*a*) and Z-project (thick slice) (*b*) of Willow C specimen after the final shear displacement step. A diamond-shaped shear zone is observed as highlighted in the shear strain ($\epsilon_{xz}$) (*c*) and *x*-displacement ($U_x$) (*d*) DVC data. (Online version in colour.)

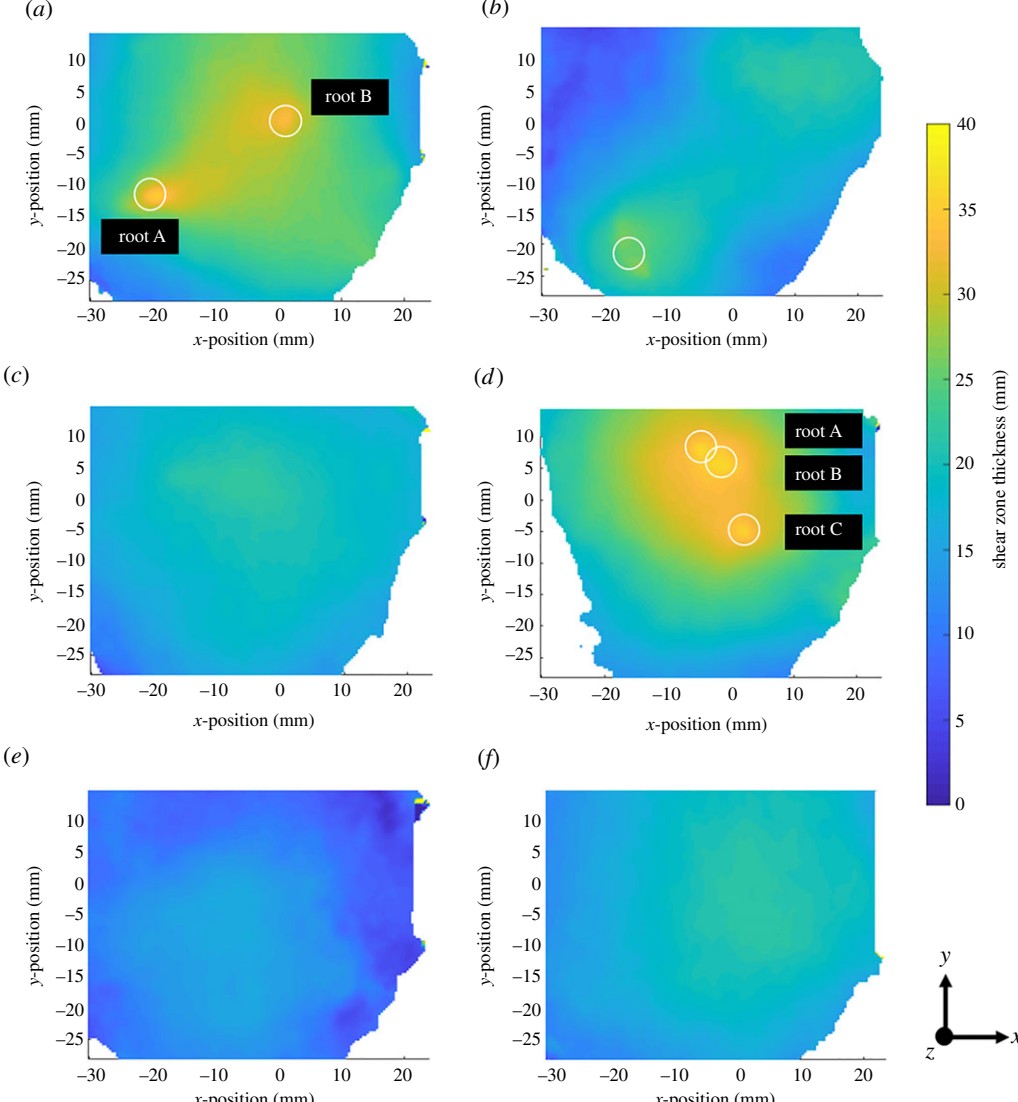

**Figure 7.** Shear zone thickness maps at the final load step for each specimen. Roots greater than 1 mm in diameter are circled. (*a*) Willow C; (*b*) Willow F; (*c*) Gorse A; (*d*) Gorse G; (*e*) Fallow D; (*f*) Fallow P. (Online version in colour.)

which the shear zone thickness was found to be constrained at the sidewalls (by the discrete discontinuities that form in the soil between the two halves of the specimen) but then extend in height into the centre of the specimen. As the thickness of the shear zone affects the magnitude of tension developed in roots (equation 2.1), it is important to understand the non-uniform shear zone formed in direct shear conditions within the cylindrical specimens.

Local shear zone maps were generated for each specimen at the final load step, using the automated procedure described earlier. Figure 4 shows the region of interest (dotted line) sampled by the local shear zone maps shown in figure 7. Roots greater than 1.0 mm diameter are circled. The region of interest was local owing to the local nature of the XCT scans and features moving out of the field of view during direct shear, which further reduced the region of interest captured by DVC.

The shear zone thickness maps (figure 7) show non-uniformity in all specimens tested. In three dimensions, the shear zone forms the shape of a slightly distorted double cone (two cones placed base to base) so that the shear zone is thickest near the centre of the tube, and smallest near the

sidewalls (giving the diamond shape when viewed in section in figure 6). Direct shear of the two cylindrical specimen halves gives a relative $x$-displacement around the full perimeter of the specimen, constraining the shear band height at the perimeter to approximately the vertical gap between the two tubes (of 2.0 mm).

Figure 7 shows that the presence of roots greater than 1.0 mm in diameter led to further localized increases in shear zone thickness, which can be seen by comparing the positions circled (in figures 4 and 7) with the nearby regions away from the root. In addition to local behaviour near roots, figure 7 also shows that globally the shear zone thickness is greater in rooted than in unrooted specimens. This agrees with findings by [17] and is likely to be caused by the transfer of axial root stresses into the soil above and below the plane of shearing.

The Gorse A specimen had very shallow roots, resulting in a small RAR (0.03%) at the shear plane. As a result, the response of Gorse A was very similar to the unrooted specimens; figure 7 shows that Gorse A had the same shear zone thickness characteristics as Fallow P.

Beyond non-uniformity in shear zone thickness, it is also possible that soil conditions affected the magnitude of shear zone thickness. The shear zone thickness in Fallow D was approximately 25% less than Fallow P. As discussed earlier (in §3a(i)), Fallow D contained soil peds and showed a twofold increase in maximum shear stress. Comparisons between Willow F and Willow C show a smaller overall shear zone thickness in the Willow F specimen (approx. 35% lower), which could be attributed to the drier soil conditions in Willow F.

As the measurements shown in figure 7 were obtained using an automated method, it is important to verify the observations. Figure 8 shows line plots of $x$-displacement versus $z$-position (depth) taken from the DVC results at locations on or as near as possible to the centre line of the segmented roots, from an $x$–$y$ position about 5 mm away from segmented roots in the direction of the centre of the tube (marked on the plot as 'away from root'), from the centre of the tube (fallow specimens) and from the edge of the DVC field of view (which did not extend to the edge of the specimen; figure 4). Visual verification can be undertaken by assessing the depth between the two knee points on the plot. The clearest difference can be seen in regions towards the edge of the DVC field of view (green lines), which have a shallower shear zone thickness than in regions at the centre (red lines). Comparing the traces at the roots (blue lines) and away from the roots (red lines), the shear zone thickness near the roots may be seen to be greater.

### (ii) Development of shear zone with shear displacements

Analytical models typically do not consider how the shear zone thickness changes with increasing shear displacement; often, the shear zone thickness is assumed to remain constant [22,23,41]. Figure 9 shows a map of the difference in shear zone thickness between the last displacement step (step 7) and displacement step 3 (the first displacement step with a clearly defined shear zone). There is a clear difference in shear zone growth between root-reinforced and unrooted specimens. Unrooted specimens show a reduction in shear zone thickness with shear displacement; rooted specimens Willow C and Gorse G exhibited shear zone thickness growth; Willow F showed a constant shear zone thickness. This is evident in figure 10 where local measurements taken at roots on Willow C and Gorse G showed a localized increase in shear zone thickness approaching 20% between displacement steps 3 and 7. The mechanism of shear zone thickness growth in rooted specimens appears to be the transfer of (increasing) axial root stresses into the soil above and below the shear plane through friction and anchorage, causing a localized widening of the shear zone.

### (c) Root stresses and soil–root interaction

### (i) Measurement of root extension and comparison with an analytical model

In analytical approaches such as the Waldron model, the average axial tensile stress acting on a root $\sigma_{axial}$ depends on the elastic (Young's) modulus of the root $E$, the change in length of the root

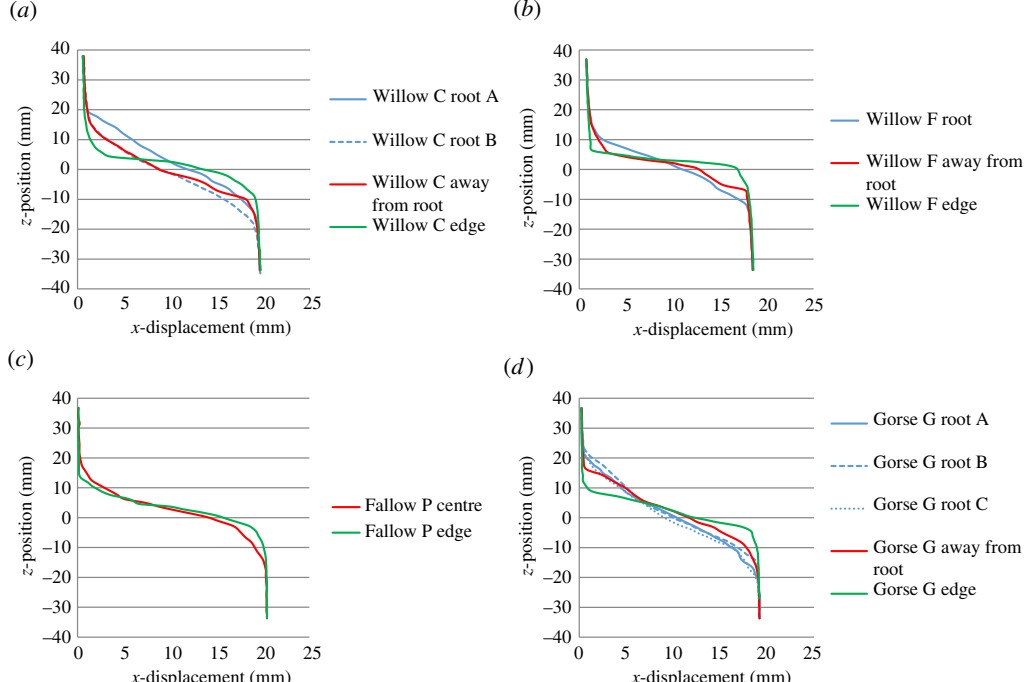

**Figure 8.** Depth versus $x$-displacement line profiles comparing responses at the root (blue lines), about 5 mm away from the root or at the centre of the specimen (red lines), and towards the edge of the DVC field of view (green lines). The displacements are determined from the DVC and are the difference between step 7 and the undeformed volume. (*a*) Willow C; (*b*) Willow F; (*c*) Fallow P; (*d*) Gorse G. (Online version in colour.)

$\Delta l$ and the initial length of the root under tension $l_0$:

$$\sigma_{\text{axial}} = \left(\frac{\Delta l}{l_0}\right) E. \tag{3.1}$$

The change in length of the root $\Delta l$ is calculated by

$$\Delta l = Z(\sec \beta - 1). \tag{3.2}$$

According to equations (3.1) and (3.2), $\sigma_{\text{axial}}$ scales linearly with $\Delta l$. Furthermore, $\Delta l$ is sensitive to the measurement of shear zone thickness $Z$ (equations (2.3) and (3.2)). It is therefore important to test the accuracy of $\Delta l$ calculated using equation (3.2). This was achieved by making comparisons with direct measurements of $\Delta l$ from XCT scan data of the root path geometry. $\Delta l$ was taken as the difference between the measured lengths of each root unloaded and at displacement step 7. Two branching regions (one above and one below the shear zone) were used as reference points to determine the original and deformed lengths of the roots. Owing to the lack of texture on the root and the resulting relatively coarse resolution, it was not possible to track local axial root deformation in more detail.

Figure 11 compares $\Delta l$ calculated using equation (3.2), the shear zone thickness and the shear displacement (solid lines) with the direct measurements (dotted lines), for increasing shear displacement. In all cases, values of $\Delta l$ derived from equation (3.2) were 10–30% greater than the difference between the measured length at the final displacement step. This is because the calculation using equation (3.2) does not consider the exact path of the root, which is shown in figure 12 to locally compress the soil, forming a radius in bending. This can be seen in figure 13 where the path taken by the root is shorter than the tri-linear fit (the basis of equation (3.2)) used in the Waldron model. The shorter path actually taken by the root leads to a lower $\Delta l$ than calculated

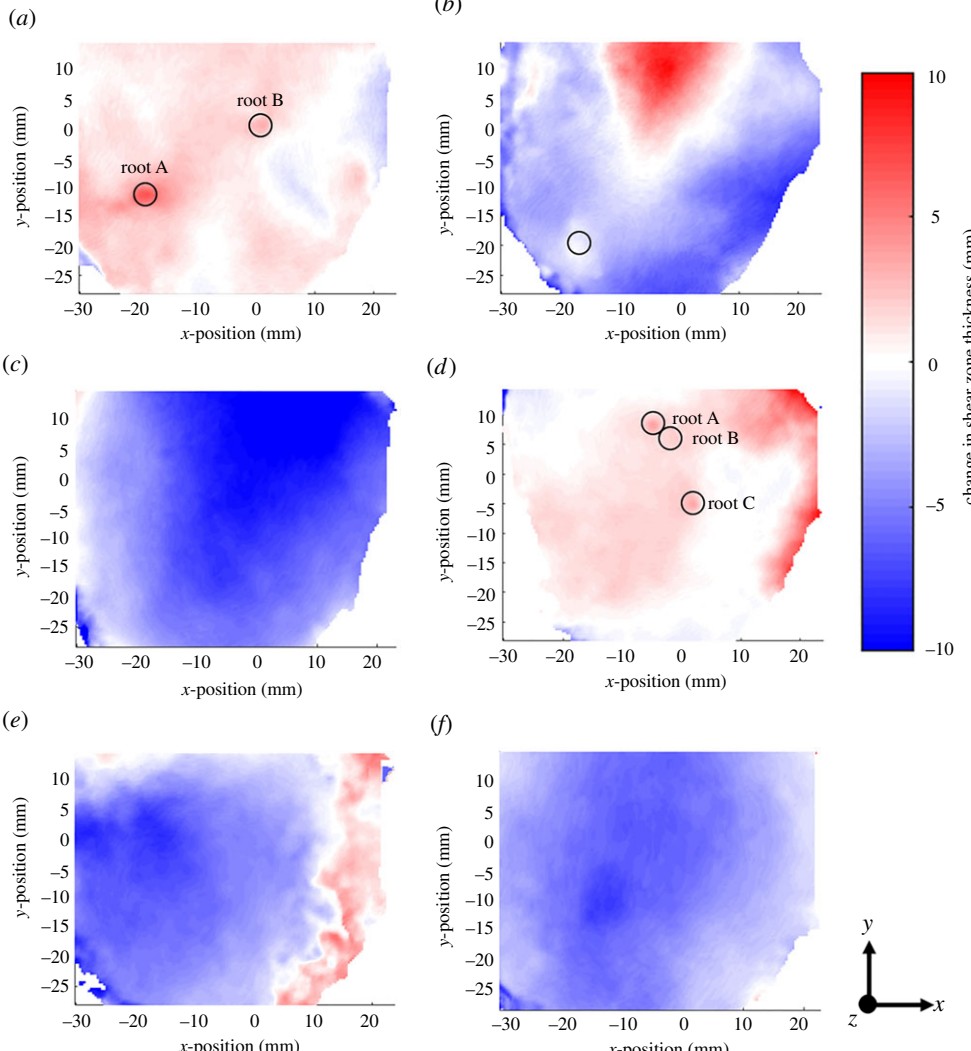

**Figure 9.** Change in shear zone thickness between displacement steps 3 and 7. Positive values represent an increase in shear zone thickness with increasing shear displacement. Roots greater than 1 mm in diameter are circled. (*a*) Willow C; (*b*) Willow F; (*c*) Gorse A; (*d*) Gorse G; (*e*) Fallow D; (*f*) Fallow P. (Online version in colour.)

using equation (3.2) and will therefore result in smaller root stresses, hence less reinforcement effect, for a given shear displacement.

### (ii) Determining dominant soil–root interface mechanisms: experiments using analogue roots

To be able to carry an axial stress across the shear zone, a root needs to be held at the ends. This could be achieved either by a soil–root interface shear stress acting on the lengths of root on either side of the shear zone (whose limiting value is defined by the parameter $\tau'$), or by the anchoring effect of branching and lateral roots. The analogue roots may represent the two extreme conditions; the unanchored ABS fibre in Analogue M is likely to be smoother than a real non-branching root, and the aluminium discs linked by single and double fibres (Analogue Q and R, respectively) likely create a near-perfect anchoring between the root and soil.

Figure 14 shows the results of the artificial root analogue tests. The smooth artificial root (Analogue M) experienced pull-out, which led to only a minor increase in shear stress compared

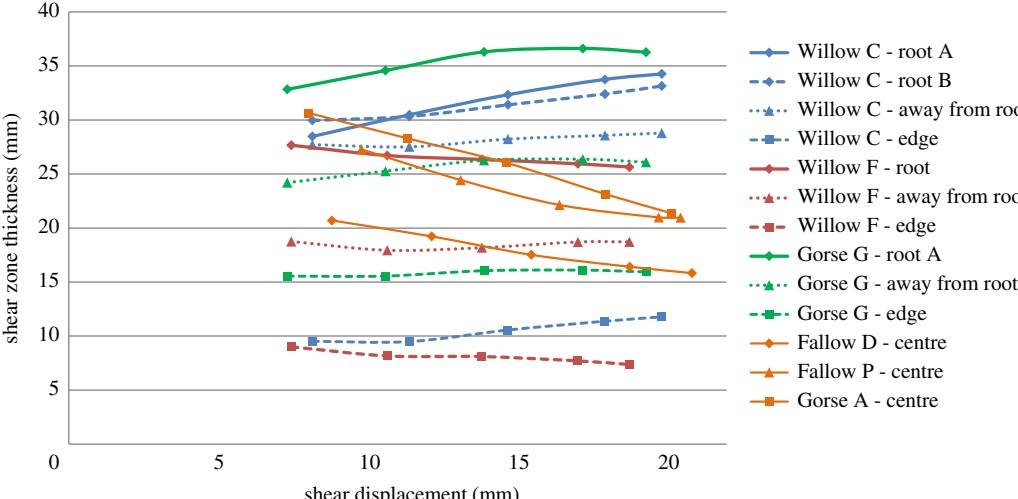

**Figure 10.** Change in shear zone thickness against shear displacement at different positions within rooted and unrooted test specimens. The results are for locations on or as near as possible to the segmented roots, from an x-y position about 5 mm away from segmented roots in the direction of the centre of the tube (marked on the plot as 'away from root'), from the centre of the tube (fallow specimens) and from the edge of the DVC field of view (which did not extend to the edge of the specimen; figure 4). (Online version in colour.)

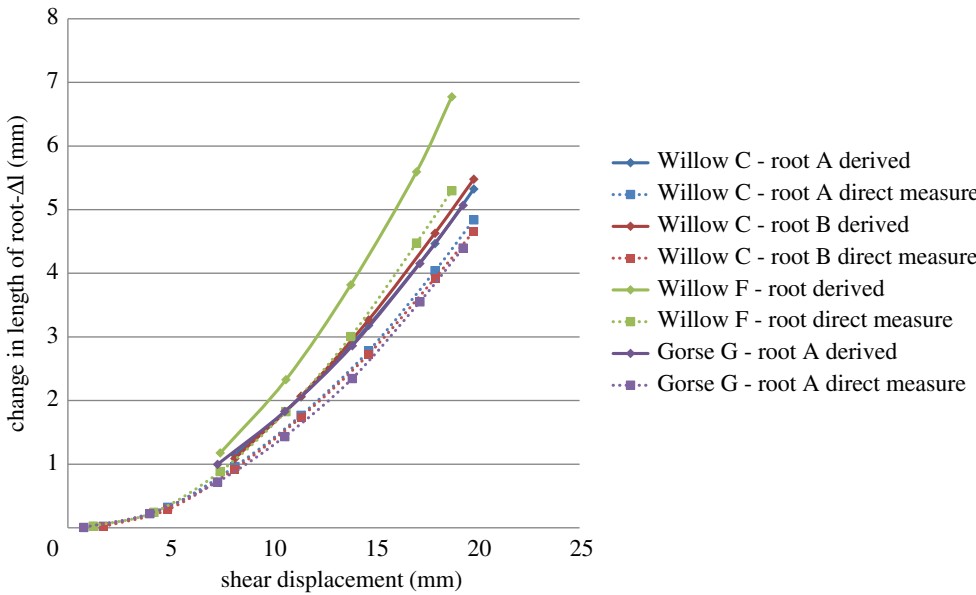

**Figure 11.** Comparison between changes in root length derived using equation (3.2) and measured directly from the root path geometry. (Online version in colour.)

with the unrooted specimen. At shear displacements greater than 12 mm, hardening behaviour is observed. This might be attributable to a localized increase in normal stresses (hence friction) between the artificial root and soil as the root in bending compresses the soil locally (figure 12).

Anchoring the artificial roots at their ends (Analogue Q and R with single and double fibres, respectively) led to a large increase in shear stress compared with the unanchored root, and a

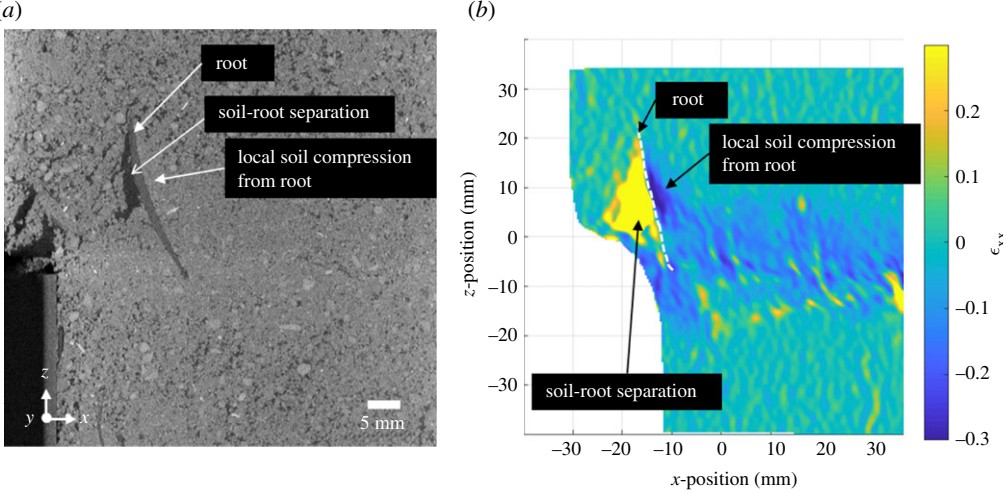

**Figure 12.** Local soil–root behaviour in the region where root bending is observed. Local features are illustrated on the XCT cross section (*a*) and DVC *x*-direction axial strain ($\epsilon_{xx}$) map (*b*). (Online version in colour.)

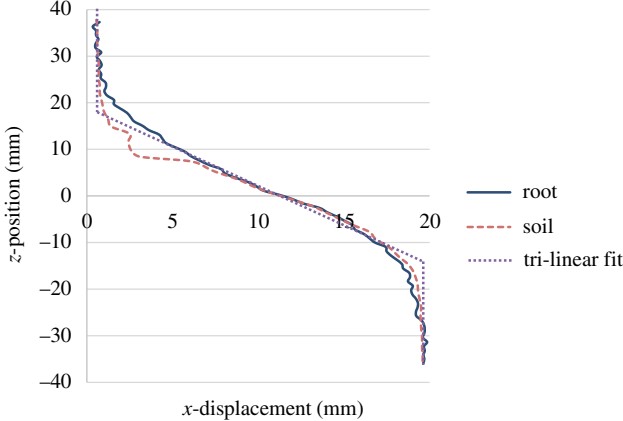

**Figure 13.** Plot showing *x*-displacement paths through the depth of the soil column (*z*-position). Different paths are observed between direct measurement of the root, DVC measurement of the soil near the root, and the tri-linear fit. (Online version in colour.)

shear stress–displacement behaviour more consistent with the real rooted specimens (figure 3). This demonstrates that anchoring by branches and lateral roots is likely to be a more important mechanism in the mobilization of real plant roots than simple friction between the root and soil. Doubling the number of anchored roots doubles the reinforcement effect: this is consistent with the linear scaling in $\Delta S$ with RAR $a_r$ in equation (2.1).

## (d) Root reinforcement predictions using the Waldron model

### (i) Waldron model sensitivity study

The Waldron model (equation (2.1) to equation (2.3)) uses a number of input parameters to establish the additional shear resistance provided by the roots. Of these, the RAR and root diameters were obtained directly from macroscopic measurements at the shear plane. The root stiffness (willow approximately 200 MPa, gorse approximately 500 MPa) and soil friction angle (36°) were determined experimentally in a previous study [9].

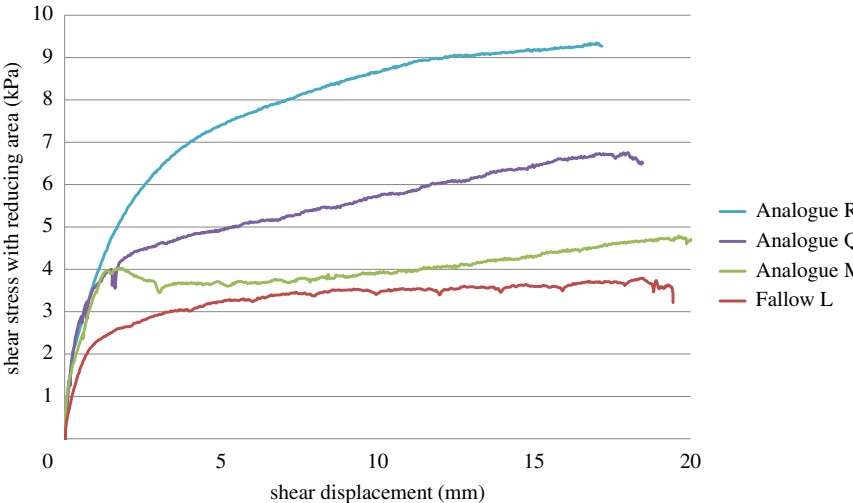

**Figure 14.** Shear stress–displacement curves for specimens containing an artificial root and a fallow specimen. (Online version in colour.)

The model assumes that

— The roots cross the shear zone perpendicular to it (in the undeformed state).
— Roots are linear elastic, with Young's modulus $E$.
— Roots never break in tension.
— There is always enough root length available to transfer the entire root tensile stress into the surrounding soil through root–soil interface friction, i.e. there is no root pull-out.

As discussed earlier, the shear zone thickness is in reality spatially non-uniform. Considering many past studies have taken the shear zone thickness as constant, it is important to perform a sensitivity study on this parameter to identify the range in estimates of additional shear resistance provided by the roots. Similarly, the parameter $\tau'$ is difficult to measure, with previous studies using empirical fits to shear test measurements to gain an approximate value [22].

Figure 15 indicates the sensitivity of the additional soil resistance provided by the roots ($\Delta S$), calculated using the Waldron model, to the shear zone thickness ($Z$) and the interface friction stress between the root and soil ($\tau'$). In figure 15$a$, the shear zone thickness values considered range from 2 mm (the thickness of the gap between the upper and lower portions of the tubes) to 30 mm (the upper limit determined from the DVC study). $\tau'$ was fixed to 1 kPa to observe $\Delta S$ sensitivity with varying $Z$. The plots indicate that the additional soil shear resistance provided by roots increases fastest at smaller shear zone thicknesses. This may be expected, as root elongation (strain) develops more rapidly for a root spanning a thinner, rather than a thicker, shear zone.

The effect of $\tau'$ on $\Delta S$ was investigated over a range of values 0.5 kPa $< \tau' <$ 5 kPa, representing different degrees of soil–root interconnectivity (i.e. representing roots with few to many branches; [22]), and as the soil is a frictional material, increasing normal and shear soil–root interface stresses with depth below the ground surface. $\Delta S$ is most sensitive to $\tau'$ when the shear zone thickness is small (figure 15$b$).

The sensitivity of the length of roots under stress, $l$, calculated by the Waldron model was also investigated:

$$l = \frac{((4\tau'ZE/D)^{0.5}(\sec\beta - 1)^{0.5}D)}{(2\tau')}. \tag{3.3}$$

Figure 15$c$ shows how the calculated length of root under stress varies with $\tau'$ (0.5 kPa $< \tau' <$ 5 kPa, as before) for two shear zone thicknesses. The root diameter $D$ was set to

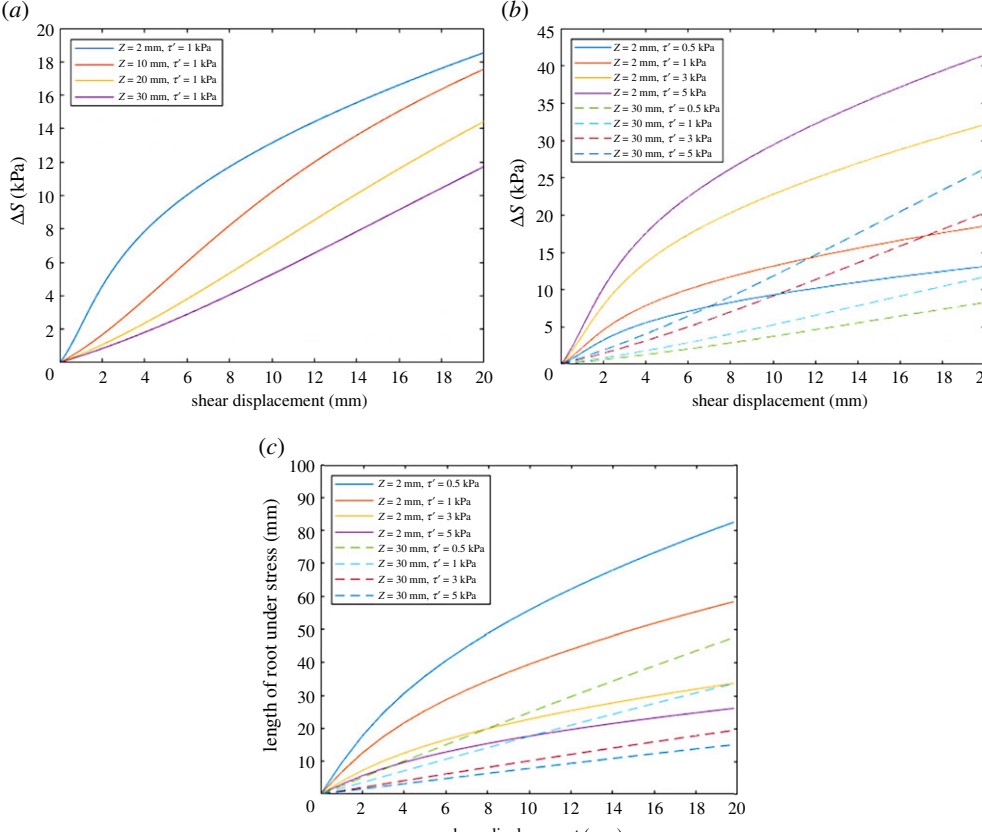

**Figure 15.** Waldron model parameter sensitivity study. (*a*) Effect of shear zone thickness ($Z$) on the additional soil resistance provided by the roots ($\Delta S$). (*b*) Effect of soil–root interconnectivity ($\tau'$) and shear zone thickness on $\Delta S$. (*c*) Effective length of a 1.0 mm diameter root under stress for different values of $\tau'$ and $Z = 2$ or $Z = 30$ mm. (Online version in colour.)

1.0 mm. The total root length under stress was typically less than 100 mm. This is consistent with the XCT observations of roots and experimental results, which do not show any sudden decrease in shear stress as would be associated with roots pulling out or breaking.

## (ii) Adapting the Waldron model to better predict root reinforcement effects

The results presented earlier show that the shear zone thickness varies within the cross section of the tube. The size of the shear zone will influence the level of stress developed in a root and is dependent on the position of the root within the tube. An expression was developed (equation (3.4)) to estimate the local shear zone thickness, based on the position of each individual root:

$$Z_{\text{root}} = Z_{\text{max}} - d_{\text{root}} \left[ \frac{Z_{\text{max}} - Z_{\text{min}}}{D_{\text{tube}}/2} \right], \tag{3.4}$$

where $Z_{\text{root}}$ is the local shear zone thickness at a given root, $d_{\text{root}}$ is the distance of the root from the centre of the tube and $D_{\text{tube}}$ is the tube diameter. $Z_{\text{min}}$ and $Z_{\text{max}}$ are the minimum and maximum shear zone thicknesses, respectively, with $Z_{\text{min}} = 2$ mm (i.e. the gap between the lower and upper tubes), and $Z_{\text{max}}$ set to the maximum shear zone thickness determined from the shear zone thickness map (figure 7).

A further consideration is the observed growth in shear zone thickness with increasing shear deformation in root-reinforced specimens. To include this behaviour in the Waldron model, the shear zone thickness was scaled linearly from 0 to 80% of the peak shear zone thickness

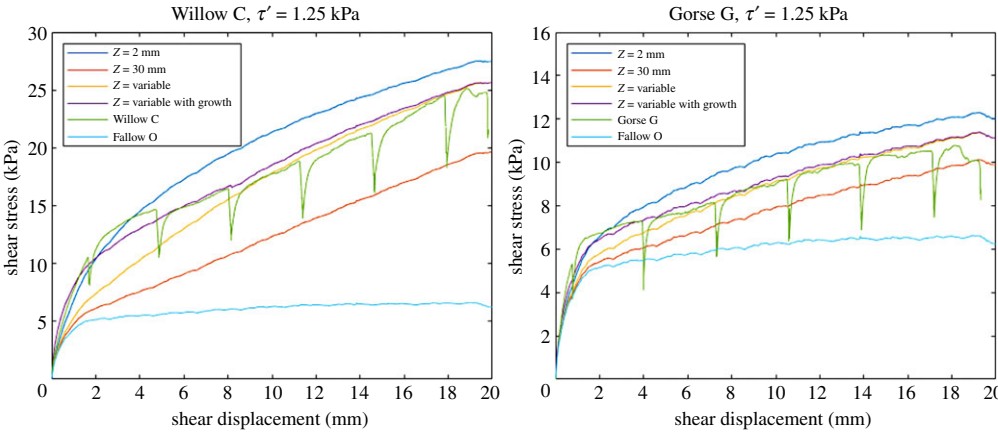

**Figure 16.** Shear stress versus shear displacement for Willow C and Gorse G specimens. Plots show modified Waldron model ($Z = 2$ mm, 30 mm, spatially variable and variable with shear zone thickness growth) and experimental results (Willow C, Gorse G and Fallow O). In the $Z =$ variable case, the shear zone thickness $Z$ is determined from the position of the root on the shear plane. (Online version in colour.)

between 0 mm shear displacement and the third displacement step (where the shear zone was first observed), then from 80 to 100% of the peak shear zone thickness between the third and final displacement steps, to approximate the measured growth of the shear band thickness shown in figure 10.

Figure 16 compares the shear stress versus shear displacement profiles calculated using various interpretations of the Waldron model (constant shear zone thicknesses of 2 and 30 mm; shear zone thickness varying with the position of root but not with shear displacement; and shear zone thickness varying with both the position of root and with shear displacement) with the experimental data for a willow (Willow C) and a gorse (Gorse G) specimen. A value of $\tau' = 1.25$ kPa was used so that the variable height shear band models replicated the final (at 20 mm) measured shear stress.

Adaptation of the Waldron model to include the variable thickness of the shear zone observed in direct shear tests gives a closer fit to the experimental data than using a fixed shear zone thickness, particularly at larger displacements. The closest agreement between the calculated and experimental results, particularly in the shape of the curve obtained, was for the model that included variability in the shear zone thickness both spatially and with shear displacement. Accounting for the spatial variation in shear zone thickness but not its growth with shear displacements leads to the underestimation of the root reinforcing effect at shear displacements less than about 8 mm.

In these analyses, the peak stresses in roots were found to be less than 80% of the root tensile strength [9] and the length of roots under stress less than 70 mm, at 20 mm applied shear displacement. Hence both root failure and pull-out were not considered in the analyses.

A back analysis using the adapted Waldron model (i.e. shear zone thickness varying with both the position of the root and shear displacement) of the analogue tests with the roots that are effectively anchored above and below the shear plane (Analogue Q and Analogue R) gives approximately $\tau' = 2.25$ kPa. Consideration of $\tau'$ based purely on friction (equal to $0.5 \times \tan\varphi'$) at the soil–root interface at the depth of the shear plane gives $\tau' = 1.0$ kPa. The value required to represent the willow and gorse root–soil interface resistance ($\tau' = 1.25$ kPa) is between these values; the real roots will provide some frictional resistance and anchoring, but this is unlikely to be as effective as the aluminium discs attached to the root analogues. It is interesting that analysis of the results from both the willow and gorse give the same value of $\tau'$; the root architectures are not the same (see [9]), although both do have a smaller number of larger diameter vertical roots

with many smaller lateral roots. The pot-grown root system may also affect the development of branching and laterals that help to anchor the roots.

## 4. Discussion

A key parameter in models of root–soil interactions is the shear zone thickness, which governs the tensile stress developed within a root when a soil containing roots is subjected to direct shear. The development of tensile stresses increases the shear resistance of the material, both directly and by increasing normal stresses (and hence frictional resistance) in the soil. A significant challenge in quantifying the behaviour of rooted soil has been understanding the behaviour of the shear zone and how it develops in the presence of roots [17]. XCT and DVC have shown that the shear zone in direct shear experiments varies in thickness, both spatially depending on the proximity of roots and specimen boundaries and also with relative displacement across the shear plane. The form of the shear zone in direct shear is considered in the soil mechanics literature [38–40], but has not been widely demonstrated experimentally.

Rooted specimens exhibited generally greater shear zone thickness and growth than unrooted specimens. Furthermore, the shear zone was shown to be locally thicker in the proximity of roots. It is hypothesized that the growth of the thickness of shear zone is due to the transfer of stress from the root into the soil, over a stressed root length that increases (along with the root axial stress) during shear.

Shear stress versus displacement relationships for rooted soils calculated using a Waldron model modified to account for variation in the thickness of the shear zone both spatially and with increasing shear displacement, closely matched experimental measurements. Other assumptions in the Waldron model are supported either directly or by inference by the experiments undertaken in this study. These include the linear relationship between root area fraction and shear resistance, and an increase in shear stress with shear displacement for a fitted value of $\tau'$ [22]. Measured changes in deformed root length were, however, overestimated by the Waldron model, by 10–30%. This was attributed to the idealized tri-linear root deformation assumed in the model and the consequent neglect of root bending at the edges of the shear zone. The overestimation of root deformation may have ramifications for the accurate estimation of the stresses developed in the root in the later stages of shear. Future studies could better estimate the level of stress in the root by considering the compression of the soil locally at the root bending points.

The most uncertain parameter in the model is the limiting root–soil interface shear stress $\tau'$, which has to be determined by curve fitting to the experimental results [22,37]. This led to further questions regarding the mechanisms preventing pull-out of the root on either side of the shear zone, i.e. pure friction at the soil–root interface, or anchorage by branching roots. Tests on root analogues suggested that purely frictional mechanisms provide little gain in resistance beyond that established at the beginning of the shear test, although the shear resistance increased a little with root angular rotation. Bending of the roots at the edges of the shear zone causes a local lateral compression of the soil, increasing the normal stress and hence friction at the soil–root interface, which could explain the increase in shear resistance towards the end of the test. The fully anchored root analogues (representative of extensive root branching) gave much larger increases in shear resistance with applied displacement. This raises questions concerning the optimal level of branching along with the root length, and how branching could be captured in the parameter $\tau'$ or through other approaches in the model. In practice, the typical plant root architecture will mean that root anchorage by branching and lateral roots is likely to be greater above the shearing plane than below it [42]. Changes in limiting interface shear stress due to both root bending and increasing normal effective stress with depth within the soil [37] would also affect the root–soil interface stress profile along the length of the root. These all provide further modelling challenges.

In summary, the modified Waldron model captured the overall shear stress versus shear displacement profile reasonably well when non-uniformity in shear zone thickness and growth were included. The next stage would be to investigate the suitability of such models for larger

scale problems, in particular including vegetation reinforcement effects in slope stability. The key questions concern the size, shape and growth of the shear zone at larger scales (e.g. [26]), and the effects of more realistic plant rooting behaviour. The experiments here have considered roots and analogues that are sub-vertical and normal to the prescribed shear surface; in the field the shear surface may be angled, and plant roots will extend outwards to meet the shear surface at a range of angles relative to the direction of shear [43,44]. In a slope, plants also often have an asymmetric root architecture in order to distribute mechanical forces to best support the mass of the plant [42].

## 5. Conclusion

An experimental study using XCT and DVC has been carried out to better understand the behaviour of root reinforcement in soils in direct shear. The direct shear tests were complemented by detailed information on root size and root distribution, and tests on artificial root analogues. The experimental data were used to assess the predictive capability of a Waldron-type model for the shear resistance of root-reinforced soils. It was found that

1. The presence of roots led to an increase in the shear zone thickness compared with fallow (unrooted) soil. In rooted soils, the thickness of the shear zone was found to increase during direct shear, in contrast with unrooted specimens where the shear zone reduced in thickness as shear progresses. Furthermore, at each stage of the test, the shear zone was smallest at the specimen boundaries (walls) and thickest close to the centre of the tube.
2. When both of these features were included in the Waldron model, measured shear stress versus shear displacement profiles were reasonably reproduced.
3. The idealized tri-linear root deformation assumed in the Waldron model overestimated the true deformed root length, as imaged in the experiments, by 10–30%. The overestimation of root deformation may affect the accurate estimation of the stresses developed in the root in the later stages of shear.
4. Root analogue tests were carried out to understand how the lengths of root on either side of the shear zone were restrained from slippage, i.e. mainly by friction at the soil–root interface or through root branching and anchorage. Root analogues relying only on friction exhibited a small increase in shear resistance and were relatively easily pulled through the soil. However, the shear resistance increased towards the end of the test as the angle of rotation of the root section crossing the shear zone increased. Root bending at the edges of the shear zone laterally compressed the soil locally, increasing the normal stress on the soil–root interface and hence the frictional resistance at the turning points of the root. Generally, however, anchoring by branching roots is much more effective than interface friction at holding the root within the zones of soil on either side of the shear zone.

The applicability of the modified Waldron model at slope scale remains to be tested. In addition to developing a suitable method of analysis, work is needed to understand whether plant rooting and shear zone behaviour seen in the smaller scale laboratory direct shear tests can be applied to the larger scale.

Data accessibility. Data used in this study have been made available on the Zenodo digital repository, as follows: Bull, D.J., Smethurst, J.A., Meijer, G.J., Sinclair, I., Pierron, F., Roose, T., Powrie, W. & Bengough, A.G. 2020 Dataset for effects of shear zone thickness in root-reinforced soils: Willow C X-ray CT scans and DVC data. Zenodo Digital Repository. (doi:10.5281/zenodo.3556741) [45]. Bull, D.J., Smethurst, J.A., Meijer, G.J., Sinclair, I., Pierron, F., Roose, T., Powrie, W. & Bengough, A.G. 2020 Dataset for effects of shear zone thickness in root-reinforced soils: Willow F X-ray CT scans and DVC data. Zenodo Digital Repository. (doi:10.5281/zenodo.3556792) [46]. Bull, D.J., Smethurst, J.A., Meijer, G.J., Sinclair, I., Pierron, F., Roose, T., Powrie, W. & Bengough, A.G. 2020 Dataset for effects of shear zone thickness in root-reinforced soils: Gorse A X-ray CT scans and DVC data. Zenodo Digital Repository. (doi:10.5281/zenodo.3556805) [47]. Bull, D.J., Smethurst, J.A., Meijer, G.J., Sinclair, I., Pierron, F., Roose, T., Powrie, W. & Bengough, A.G. 2020 Dataset for

effects of shear zone thickness in root-reinforced soils: Gorse G X-ray CT scans and DVC data. Zenodo Digital Repository. (doi:10.5281/zenodo.3556824) [48]. Bull, D.J., Smethurst, J.A., Meijer, G.J., Sinclair, I., Pierron, F., Roose, T., Powrie, W. & Bengough, A.G. 2020 Dataset for effects of shear zone thickness in root-reinforced soils: Fallow D X-ray CT scans and DVC data. Zenodo Digital Repository. (doi:10.5281/zenodo.3558069) [49]. Bull, D.J., Smethurst, J.A., Meijer, G.J., Sinclair, I., Pierron, F., Roose, T., Powrie, W. & Bengough, A.G. 2020 Dataset for effects of shear zone thickness in root-reinforced soils: Fallow P X-ray CT scans and DVC data. Zenodo Digital Repository. (doi:10.5281/zenodo.3558539) [50]. Bull, D.J., Smethurst, J.A., Meijer, G.J., Sinclair, I., Pierron, F., Roose, T., Powrie, W. & Bengough, A.G. 2020 Dataset for effects of shear zone thickness in root-reinforced soils. Zenodo Digital Repository. (doi:10.5281/zenodo.3558554) [51].

Authors' contributions. D.J.B.: conceptualization, data curation, formal analysis, investigation, methodology, validation, visualization, writing—original draft and writing—review and editing; J.A.S.: conceptualization, formal analysis, funding acquisition, methodology, project administration, supervision and writing—review and editing; G.J.M.: formal analysis, methodology, writing—original draft and writing—review and editing; I.S.: conceptualization, funding acquisition, methodology, supervision and writing—review and editing; F.P.: conceptualization, funding acquisition, methodology, supervision and writing—review and editing; T.R.: conceptualization, funding acquisition, methodology, supervision and writing—review and editing; W.P.: conceptualization, funding acquisition, methodology, supervision and writing—review and editing; G.B.: conceptualization, funding acquisition, methodology, supervision and writing—review and editing All authors gave final approval for publication and agreed to be held accountable for the work performed therein.
Competing interests. We declare we have no competing interests.
Funding. This research was funded by the UK Engineering and Physical Sciences Research Council grant nos EP/M020177/1 and EP/M020355/1, as part of a collaboration between the University of Southampton, University of Dundee, University of Aberdeen, Durham University and the James Hutton Institute. The James Hutton Institute receives funding from the Scottish Government (Rural & Environmental Services & Analytical Services Division). The authors acknowledge the μ-VIS X-ray Imaging Centre at the University of Southampton for provision of tomographic imaging facilities, supported by EPSRC grant no. EP/H01506X.
Acknowledgements. Dr Sonja Schmidt carried out the initial development work on the shear rig, with technical support from Harvey Skinner and Karl Scammell.

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
