## [Peer Review File · Proceedings. Mathematical, Physical, and Engineering Sciences]

Review History

RSPA-2021-0210.R0 (Original submission)

Review form: Referee 1

Is the manuscript an original and important contribution to its field?

Good

Is the paper of sufficient general interest?

Good

Is the overall quality of the paper suitable?

Good

Can the paper be shortened without overall detriment to the main message?

No

Do you think some of the material would be more appropriate as an electronic appendix?

Yes

Do you have any ethical concerns with this paper?

No

Recommendation?

Major revision is needed (please make suggestions in comments)

Comments to the Author(s)

Comments on "Modelling of stress transfer in root reinforced soils informed by 4D X-ray computed tomography and digital volume correlation data"

This manuscript presents a study of the changes in shear resistance and soil deformations during direct shear due to roots using the non-destructive technique of X-ray CT and digital volume correlation. The authors carried out 3D analysis of local displacements and strains on root-reinforced soils and unrooted. Conventional measurements are also made to complement CT data. The subject of the paper is interesting for the audience of Proceedings A. The paper has interesting results, but before its acceptance for publication, the authors need to clarify some points.

I suggest including some more results in the abstract. The authors only presented a general overview of the results in the abstract, but I believe that some more specific results could be interesting for the readers. Please, insert a last sentence in the abstract highlighting the novelty of the results obtained.

The introduction is well written, however, my suggestion is to move the theoretical description of analytical models such as the paragraphs containing equations and figures to a new section, perhaps a theory section? Or simply to incorporate this information in the methodology.

The methodology needs some clarification. Please, give more details about the soil preparation in the tubes. Do the authors believe that only two replications are enough for the CT studies? What was the purpose of the matric potential measurements in the CT? How to know that the matric potential was the same across the entire sample? Any influence of the step differences presented in Table 2 in the CT results among treatments? Was the voxel resolution obtained enough for the study proposed? Please, insert a reference for the DaVis LaVision software. As the digital volume correlation is one of the techniques utilized in the paper to support the author's findings; I suggest including the description of the methodological steps followed in the paper rather than making reference to another paper. How was determined the two knee points on the tri-linear profile? To me, the procedure presented in Fig. 3 seemed a little bit subjective?

In my option, there is an excessive number of figures in the paper, twenty! I suggest to the authors to merge some of them, at least those dealing with similar findings. The same comment is worthy for the tables. Perhaps, some of the figures and tables would included as supplementary material.

What is the explanation for the discontinuities shown in the behavior of the shear stress curves for some of the treatments presented in Fig. 4? Why some treatments presented this behavior and others do not? Were the differences in the soil bulk density, matric suction, and water content significant among treatments? Any possible effect of the not resolved roots by CT in the results observed?

If possible, I suggest to the authors merge the results and discussion sections. Some of the results presented were not covered in the discussion. I suggest to the authors to improve them a little bit in the discussion section. Perhaps, it is necessary to work a little bit more on it to make clear the relationships among all the results presented in the study.

Overall evaluation: for the reasons stated above, the manuscript needs additional revision.

Review form: Referee 2

Is the manuscript an original and important contribution to its field?

Excellent

Is the paper of sufficient general interest?

Excellent

Is the overall quality of the paper suitable?

Good

Can the paper be shortened without overall detriment to the main message?

No

Do you think some of the material would be more appropriate as an electronic appendix?

Yes

Do you have any ethical concerns with this paper?

No

Recommendation?

Accept with minor revision (please list in comments)

Comments to the Author(s)

In this work, the authors evaluate shear resistance from root structures in root reinforced soils by X-ray computed tomography and digital volume correlation. Previous studies investigating the mechanisms of soil-root reinforcement focus on behavior at boundaries due to experimental challenges. This study uses a novel approach developed recently by the authors (ref 21) that uses X-ray CT and DVC to investigate shearing and deformations in bulk soil. Three types of specimen were tested: willow, gorse, and fallow. A strong correlation between increased root area ratio in the shear zone and increased shear stress was revealed. Analysis by DVC indicates shows a non-uniform shear zone within the cylinder of soil; instead, shear zone thickness tapers off towards the edges. Roots greater than 1 mm in diameter lead to an observable local increase in shear zone thickness. To better understand the influence of lateral roots in anchoring the root, the authors utilize ABS plastic filament and aluminum discs to simulate non-branching and branching roots. The anchored filament showed a shear stress response curve similar to the rooted specimens suggesting that anchoring plays an important role in soil-root reinforcement. Introduction of variability in shear zone thickness by space and with shear displacement improves predictions of shear stress response by the Waldron model.

Major comments:

The work presents a clear advance by showing how X-ray CT and DVM experiments can be used to refine and improve models of root-reinforced soil shearing. However, important methodological details were published in reference 21 but not summarized here in enough detail, making it difficult to follow the results and understand the data by only reading this manuscript (for example, nowhere in the manuscript I can find the explanation of ϵ_{xz} and ϵ_{xx}). It appears some of the figures were re-used - top panel of Figure 8, left panel of Figure 15 (the examples of CT images) are the same as Figure 13 and 15 in ref. 21. Overall, it would benefit the reader to have some material, such as the variation in fallow tests and root count results sections as well as some figures, moved to supplemental information to streamline the narrative.

Given the relatively high-resolution of the CT scanning, it was puzzling why these data weren't used to greater effect to measure the roots within and without the shear area (including even the root system architecture or other measures of connectivity). Please explain why analysis of the

roots (Figure 6) was limited to the macroscope experiment, and more details on how the measurements were taken from it. If there are major impediments to using 3D CT data, please note and discuss the potential implications of understanding shearing processes with the entire root system in mind. For example, whether it would be interesting to analyze the relationship between shear stress and other root measurements besides root area ratio.

Section 3.1.2: The authors only analyzed the x-z slice. Without comparing the 3D dimensions, how can we be confident that the peds in Fallow D are larger than in Fallow P? Page 11, Line 30-34: How can the authors make these conclusions from the results shown here? If these conclusions are from previous publications, the authors should add references.

Figure 11 and Figure 13. How many points (x, y position) were analyzed for different locations? It looks like each line represent one point on the shear plane. How was the point selected?

The authors observe roots gathered around the edges indicative of a pot-grown system. The deflected lateral roots likely increase reinforcement and are connected to the roots in the center of soil sample. Could the authors please comment on whether deflected roots at the outer boundary influence the measured root-soil interface shear stress?

It appears from Figure 1 and in testing of root analogues that there is an assumption that roots grow more or less straight down, but they often do not. What implications might this assumption have? An additional figure or diagram describing the analogue experiment would be helpful, and please consider moving these experimental details from the results to the methods section.

In the anchored filament measurements, resistance introduced by anchoring is evenly distributed above and below the shear plane. However, root systems are likely to have would have different lateral branching above and below the shear plane. How would asymmetry in anchoring/branching affect root-soil interface shear stress?

The authors acknowledge their uncertainty on whether variability in shear zone thickness would be observed in larger systems. It would benefit the reader to have this uncertainty and other differences from larger and/or sloped systems further discussed in the discussion.

In this study, shear strain is applied planar to the ground. In a sloped system, plants have asymmetric root architecture to accommodate differences in the distribution of mechanical forces into the soil. How would such an architecture affect shear stress response?

Minor comment:

Section 2.4, Page 8, Line 38: What are the several different calculation approaches?

In figure 12, please include the position of roots >1 mm in diameter as shown in figure 9 and 10.

Decision letter (RSPA-2021-0210.R0)

15-Jun-2021

Dear Dr Smethurst

The Editor of Proceedings A has now received comments from referees on the above paper and would like you to revise it in accordance with their suggestions which can be found below (not including confidential reports to the Editor).

Please submit a copy of your revised paper within four weeks - if we do not hear from you within this time then it will be assumed that the paper has been withdrawn. In exceptional circumstances, extensions may be possible if agreed with the Editorial Office in advance.

Please note that it is the editorial policy of Proceedings A to offer authors one round of revision in which to address changes requested by referees. If the revisions are not considered satisfactory by the Editor, then the paper will be rejected, and not considered further for publication by the journal. In the event that the author chooses not to address a referee's comments, and no scientific justification is included in their cover letter for this omission, it is at the discretion of the Editor whether to continue considering the manuscript.

To revise your manuscript, log into <http://mc.manuscriptcentral.com/prsa> and enter your Author Centre, where you will find your manuscript title listed under "Manuscripts with Decisions." Under "Actions," click on "Create a Revision." Your manuscript number has been appended to denote a revision.

You will be unable to make your revisions on the originally submitted version of the manuscript. Instead, revise your manuscript and upload a new version through your Author Centre.

When submitting your revised manuscript, you will be able to respond to the comments made by the referee(s) and upload a file "Response to Referees" in Step 1: "View and Respond to Decision Letter". Please use this to document how you have responded to the comments, and the adjustments you have made. In order to expedite the processing of the revised manuscript, please be as specific as possible in your response to the referee(s).

IMPORTANT: Your original files are available to you when you upload your revised manuscript. Please delete any unnecessary previous files before uploading your revised version.

When revising your paper please ensure that it remains under 28 pages long. In addition, any pages over 20 will be subject to a charge (£150 + VAT (where applicable) per page). Your paper has been ESTIMATED to be 27 pages.

Open Access

You are invited to opt for open access, our author pays publishing model. Payment of open access fees will enable your article to be made freely available via the Royal Society website as soon as it is ready for publication. For more information about open access please visit <https://royalsociety.org/journals/authors/open-access/>. The open access fee for this journal is £1700/\$2380/€2040 per article. VAT will be charged where applicable. Please note that if the corresponding author is at an institution that is part of a Read and Publishing deal you are required to select this option. See <https://royalsociety.org/journals/librarians/purchasing/read-and-publish/read-publish-agreements/> for further details.

Once again, thank you for submitting your manuscript to Proc. R. Soc. A and I look forward to receiving your revision. If you have any questions at all, please do not hesitate to get in touch.

Yours sincerely
Raminder Shergill
proceedingsa@royalsociety.org

Reviewer(s)' Comments to Author:

Referee: 1

Comments to the Author(s)

Comments on "Modelling of stress transfer in root reinforced soils informed by 4D X-ray computed tomography and digital volume correlation data"

This manuscript presents a study of the changes in shear resistance and soil deformations during direct shear due to roots using the no-destructive technique of X-ray CT and digital volume correlation. The authors carried out 3D analysis of local displacements and strains on root-reinforced soils and unrooted. Conventional measurements are also made to complement CT data. The subject of the paper is interesting for the audience of Proceedings A. The paper has interesting results, but before its acceptance for publication, the authors need to clarify some points.

I suggest including some more results in the abstract. The authors only presented a general overview of the results in the abstract, but I believe that some more specific results could be

interesting for the readers. Please, insert a last sentence in the abstract highlighting the novelty of the results obtained.

The introduction is well written, however, my suggestion is to move the theoretical description of analytical models such as the paragraphs containing equations and figures to a new section, perhaps a theory section? Or simply to incorporate this information in the methodology.

The methodology needs some clarification. Please, give more details about the soil preparation in the tubes. Do the authors believe that only two replications are enough for the CT studies? What was the purpose of the matric potential measurements in the CT? How to know that the matric potential was the same across the entire sample? Any influence of the step differences presented in Table 2 in the CT results among treatments? Was the voxel resolution obtained enough for the study proposed? Please, insert a reference for the DaVis LaVision software. As the digital volume correlation is one of the techniques utilized in the paper to support the author's findings; I suggest including the description of the methodological steps followed in the paper rather than making reference to another paper. How was determined the two knee points on the tri-linear profile? To me, the procedure presented in Fig. 3 seemed a little bit subjective?

In my option, there is an excessive number of figures in the paper, twenty! I suggest to the authors to merge some of them, at least those dealing with similar findings. The same comment is worthy for the tables. Perhaps, some of the figures and tables would included as supplementary material.

What is the explanation for the discontinuities shown in the behavior of the shear stress curves for some of the treatments presented in Fig. 4? Why some treatments presented this behavior and others do not? Were the differences in the soil bulk density, matric suction, and water content significant among treatments? Any possible effect of the not resolved roots by CT in the results observed?

If possible, I suggest to the authors merge the results and discussion sections. Some of the results presented were not covered in the discussion. I suggest to the authors to improve them a little bit in the discussion section. Perhaps, it is necessary to work a little bit more on it to make clear the relationships among all the results presented in the study.

Overall evaluation: for the reasons stated above, the manuscript needs additional revision.

Referee: 2

Comments to the Author(s)

In this work, the authors evaluate shear resistance from root structures in root reinforced soils by X-ray computed tomography and digital volume correlation. Previous studies investigating the mechanisms of soil-root reinforcement focus on behavior at boundaries due to experimental challenges. This study uses a novel approach developed recently by the authors (ref 21) that uses X-ray CT and DVC to investigate shearing and deformations in bulk soil. Three types of specimen were tested: willow, gorse, and fallow. A strong correlation between increased root area ratio in the shear zone and increased shear stress was revealed. Analysis by DVC indicates shows a non-uniform shear zone within the cylinder of soil; instead, shear zone thickness tapers off towards the edges. Roots greater than 1 mm in diameter lead to an observable local increase in shear zone thickness. To better understand the influence of lateral roots in anchoring the root, the authors utilize ABS plastic filament and aluminum discs to simulate non-branching and branching roots. The anchored filament showed a shear stress response curve similar to the rooted specimens suggesting that anchoring plays an important role in soil-root reinforcement. Introduction of variability in shear zone thickness by space and with shear displacement improves predictions of shear stress response by the Waldron model.

Major comments:

The work presents a clear advance by showing how X-ray CT and DVM experiments can be used to refine and improve models of root-reinforced soil shearing. However, important methodological details were published in reference 21 but not summarized here in enough detail, making it difficult to follow the results and understand the data by only reading this manuscript (for example, nowhere in the manuscript I can find the explanation of ϵ_{xz} and ϵ_{xx}). It appears some of the figures were re-used - top panel of Figure 8, left panel of Figure 15 (the examples of CT images) are the same as Figure 13 and 15 in ref. 21. Overall, it would benefit the reader to have some material, such as the variation in fallow tests and root count results sections as well as some figures, moved to supplemental information to streamline the narrative.

Given the relatively high-resolution of the CT scanning, it was puzzling why these data weren't used to greater effect to measure the roots within and without the shear area (including even the root system architecture or other measures of connectivity). Please explain why analysis of the roots (Figure 6) was limited to the microscope experiment, and more details on how the measurements were taken from it. If there are major impediments to using 3D CT data, please note and discuss the potential implications of understanding shearing processes with the entire root system in mind. For example, whether it would be interesting to analyze the relationship between shear stress and other root measurements besides root area ratio.

Section 3.1.2: The authors only analyzed the x-z slice. Without comparing the 3D dimensions, how can we be confident that the peds in Fallow D are larger than in Fallow P? Page 11, Line 30-34: How can the authors make these conclusions from the results shown here? If these conclusions are from previous publications, the authors should add references.

Figure 11 and Figure 13. How many points (x, y position) were analyzed for different locations? It looks like each line represent one point on the shear plane. How was the point selected?

The authors observe roots gathered around the edges indicative of a pot-grown system. The deflected lateral roots likely increase reinforcement and are connected to the roots in the center of soil sample. Could the authors please comment on whether deflected roots at the outer boundary influence the measured root-soil interface shear stress?

It appears from Figure 1 and in testing of root analogues that there is an assumption that roots grow more or less straight down, but they often do not. What implications might this assumption have? An additional figure or diagram describing the analogue experiment would be helpful, and please consider moving these experimental details from the results to the methods section.

In the anchored filament measurements, resistance introduced by anchoring is evenly distributed above and below the shear plane. However, root systems are likely to have would have different lateral branching above and below the shear plane. How would asymmetry in anchoring/branching affect root-soil interface shear stress?

The authors acknowledge their uncertainty on whether variability in shear zone thickness would be observed in larger systems. It would benefit the reader to have this uncertainty and other differences from larger and/or sloped systems further discussed in the discussion.

In this study, shear strain is applied planar to the ground. In a sloped system, plants have asymmetric root architecture to accommodate differences in the distribution of mechanical forces into the soil. How would such an architecture affect shear stress response?

Minor comment:

Section 2.4, Page 8, Line 38: What are the several different calculation approaches?

In figure 12, please include the position of roots ≥ 1 mm in diameter as shown in figure 9 and 10.

Author's Response to Decision Letter for (RSPA-2021-0210.R0)

See Appendix A.

RSPA-2021-0210.R1 (Revision)

Review form: Referee 1

Is the manuscript an original and important contribution to its field?

Good

Is the paper of sufficient general interest?

Good

Is the overall quality of the paper suitable?

Good

Can the paper be shortened without overall detriment to the main message?

Yes

Do you think some of the material would be more appropriate as an electronic appendix?

No

Do you have any ethical concerns with this paper?

No

Recommendation?

Accept as is

Comments to the Author(s)

In my opinion the paper is now ready for publication.

Review form: Referee 2

Is the manuscript an original and important contribution to its field?

Excellent

Is the paper of sufficient general interest?

Good

Is the overall quality of the paper suitable?

Excellent

Can the paper be shortened without overall detriment to the main message?

Yes

Do you think some of the material would be more appropriate as an electronic appendix?

No

Do you have any ethical concerns with this paper?

No

Recommendation?

Accept with minor revision (please list in comments)

Comments to the Author(s)

Comments were adequately addressed. Please note some figure reference issues (e.g. page 12 line 46, page 13 line 44, page 14 line 20, page 19 line 12, page 22 line 38, and page 24 line 16), and verify the Zenodo data repositories (ref 45-51) are publicly available and correctly cited.

Decision letter (RSPA-2021-0210.R1)

16-Sep-2021

Dear Dr Smethurst,

On behalf of the Editor, I am pleased to inform you that your Manuscript RSPA-2021-0210.R1 entitled "Modelling of stress transfer in root reinforced soils informed by 4D X-ray computed tomography and digital volume correlation data" has been accepted for publication subject to minor revisions in Proceedings A. Please find the referees' comments below.

The reviewer(s) have recommended publication, but also suggest some minor revisions to your manuscript. Therefore, I invite you to respond to the reviewer(s)' comments and revise your manuscript. Please note that we have a strict upper limit of 28 pages for each paper. Please endeavour to incorporate any revisions while keeping the paper within journal limits. Please note that page charges are made on all papers longer than 20 pages. If you cannot pay these charges you must reduce your paper to 20 pages before submitting your revision. Your paper has been ESTIMATED to be 28 pages. We cannot proceed with typesetting your paper without your agreement to meet page charges in full should the paper exceed 20 pages when typeset. If you have any questions, please do get in touch.

It is a condition of publication that you submit the revised version of your manuscript within 7 days. If you do not think you will be able to meet this date please let me know in advance of the due date.

To revise your manuscript, log into <https://mc.manuscriptcentral.com/prsa> and enter your Author Centre, where you will find your manuscript title listed under "Manuscripts with Decisions." Under "Actions," click on "Create a Revision." Your manuscript number has been appended to denote a revision.

You will be unable to make your revisions on the originally submitted version of the manuscript. Instead, revise your manuscript and upload a new version through your Author Centre.

When submitting your revised manuscript, you will be able to respond to the comments made by the referee(s) and upload a file "Response to Referees" in Step 1: "View and Respond to Decision Letter". Please provide a point-by-point response to the comments raised by the reviewers and the editor(s). A thorough response to these points will help us to assess your revision quickly. You can also upload a 'tracked changes' version either as part of the 'Response to reviews' or as a 'Main document'.

IMPORTANT: Your original files are available to you when you upload your revised manuscript. Please delete any redundant files before completing the submission process.

When uploading your revised files, please make sure that you include the following as we cannot proceed without these:

- 1) A text file of the manuscript (doc, txt, rtf or tex), including the references, tables (including captions) and figure captions. Please remove any tracked changes from the text before submission. PDF files are not an accepted format for the "Main Document".
- 2) A separate electronic file of each figure (tif, eps or print-quality pdf preferred). The format should be produced directly from original creation package, or original software format.
- 3) Electronic Supplementary Material (ESM): all supplementary materials accompanying an accepted article will be treated as in their final form. Note that the Royal Society will not edit or typeset supplementary material and it will be hosted as provided. Please ensure that the supplementary material includes the paper details where possible (authors, article title, journal name). Supplementary files will be published alongside the paper on the journal website and posted on the online figshare repository (<https://figshare.com>). The heading and legend provided for each supplementary file during the submission process will be used to create the figshare page, so please ensure these are accurate and informative so that your files can be found in searches. Files on figshare will be made available approximately one week before the accompanying article so that the supplementary material can be attributed a unique DOI. Alternatively you may upload a zip folder containing all source files for your manuscript as described above with a PDF as your "Main Document". This should be the full paper as it appears when compiled from the individual files supplied in the zip folder.

Article Funder

Please ensure you fill in the Article Funder question on page 2 to ensure the correct data is collected for FundRef (<http://www.crossref.org/fundref/>).

Media summary

Please ensure you include a short non-technical summary (up to 100 words) of the key findings/importance of your paper. This will be used for to promote your work and marketing purposes (e.g. press releases). The summary should be prepared using the following guidelines:

- *Write simple English: this is intended for the general public. Please explain any essential technical terms in a short and simple manner.
- *Describe (a) the study (b) its key findings and (c) its implications.
- *State why this work is newsworthy, be concise and do not overstate (true 'breakthroughs' are a rarity).
- *Ensure that you include valid contact details for the lead author (institutional address, email address, telephone number).

Cover images

We welcome submissions of images for possible use on the cover of Proceedings A. Images should be square in dimension and please ensure that you obtain all relevant copyright permissions before submitting the image to us. If you would like to submit an image for consideration please send your image to proceedingsa@royalsociety.org

Open Access

You are invited to opt for open access, our author pays publishing model. Payment of open access fees will enable your article to be made freely available via the Royal Society website as soon as it is ready for publication. For more information about open access please visit <https://royalsociety.org/journals/authors/open-access/>. The open access fee for this journal is £1700/\$2380/€2040 per article. VAT will be charged where applicable. Please note that if the corresponding author is at an institution that is part of a Read and Publishing deal you are

required to select this option. See <https://royalsociety.org/journals/librarians/purchasing/read-and-publish/read-publish-agreements/> for further details.

Once again, thank you for submitting your manuscript to Proceedings A and I look forward to receiving your revision. If you have any questions at all, please do not hesitate to get in touch.

Best wishes
Raminder Shergill
proceedingsa@royalsociety.org
Proceedings A

Reviewer(s)' Comments to Author:
Referee: 1
Comments to the Author(s)
In my opinion the paper is now ready for publication.

Referee: 2
Comments to the Author(s)
Comments were adequately addressed. Please note some figure reference issues (e.g. page 12 line 46, page 13 line 44, page 14 line 20, page 19 line 12, page 22 line 38, and page 24 line 16), and verify the Zenodo data repositories (ref 45-51) are publicly available and correctly cited.

Decision letter (RSPA-2021-0210.R2)

30-Nov-2021

Dear Dr Smethurst

I am pleased to inform you that your manuscript entitled "Modelling of stress transfer in root reinforced soils informed by 4D X-ray computed tomography and digital volume correlation data" has been accepted in its final form for publication in Proceedings A.

Our Production Office will be in contact with you in due course. You can expect to receive a proof of your article soon. Please contact the office to let us know if you are likely to be away from e-mail in the near future. If you do not notify us and comments are not received within 5 days of sending the proof, we may publish the paper as it stands.

As a reminder, you have provided the following 'Data accessibility statement' (if applicable). Please remember to make any data sets live prior to publication, and update any links as needed when you receive a proof to check. It is good practice to also add data sets to your reference list.
Statement (if applicable): Data used in this study has been made available on the Zenodo digital repository, as follows:

Bull, D.J., Smethurst, J.A., Meijer, G.J., Sinclair, I., Pierron, F., Roose, T., Powrie, W. & Bengough, A.G. 2020 Dataset for effects of shear zone thickness in root-reinforced soils: Willow C X-ray CT scans and digital volume correlation data. Zenodo Digital Repository. (doi:10.5281/zenodo.3556741) [45]. Bull, D.J., Smethurst, J.A., Meijer, G.J., Sinclair, I., Pierron, F., Roose, T., Powrie, W. & Bengough, A.G. 2020 Dataset for effects of shear zone thickness in root-reinforced soils: Willow F X-ray CT scans and digital volume correlation data. Zenodo Digital Repository. (doi:10.5281/zenodo.3556792) [46]. Bull, D.J., Smethurst, J.A., Meijer, G.J., Sinclair, I., Pierron, F., Roose, T., Powrie, W. & Bengough, A.G. 2020 Dataset for effects of shear zone thickness in root-reinforced soils: Gorse A X-ray CT scans and digital volume correlation data. Zenodo Digital Repository. (doi:10.5281/zenodo.3556805) [47]. Bull, D.J., Smethurst, J.A., Meijer, G.J., Sinclair, I., Pierron, F., Roose, T., Powrie, W. & Bengough, A.G.

2020 Dataset for effects of shear zone thickness in root-reinforced soils: Gorse G X-ray CT scans and digital volume correlation data. Zenodo Digital Repository. (doi:10.5281/zenodo.3556824) [48]. Bull, D.J., Smethurst, J.A., Meijer, G.J., Sinclair, I., Pierron, F., Roose, T., Powrie, W. & Bengough, A.G. 2020 Dataset for effects of shear zone thickness in root-reinforced soils: Fallow D X-ray CT scans and digital volume correlation data. Zenodo Digital Repository. (doi:10.5281/zenodo.3558069) [49]. Bull, D.J., Smethurst, J.A., Meijer, G.J., Sinclair, I., Pierron, F., Roose, T., Powrie, W. & Bengough, A.G. 2020 Dataset for effects of shear zone thickness in root-reinforced soils: Fallow P X-ray CT scans and digital volume correlation data. Zenodo Digital Repository. (doi:10.5281/zenodo.3558539) [50]. Bull, D.J., Smethurst, J.A., Meijer, G.J., Sinclair, I., Pierron, F., Roose, T., Powrie, W. & Bengough, A.G. 2020 Dataset for effects of shear zone thickness in root-reinforced soils. Zenodo Digital Repository. (doi:10.5281/zenodo.3558554) [51].

Open access

You are invited to opt for open access, our author pays publishing model. Payment of open access fees will enable your article to be made freely available via the Royal Society website as soon as it is ready for publication. For more information about open access please visit <https://royalsociety.org/journals/authors/which-journal/open-access/>. The open access fee for this journal is £1700/\$2380/€2040 per article. VAT will be charged where applicable.

Note that if you have opted for open access then payment will be required before the article is published – payment instructions will follow shortly.

If you wish to opt for open access then please inform the editorial office (proceedingsa@royalsociety.org) as soon as possible.

Your article has been estimated as being 28 pages long. Our Production Office will inform you of the exact length at the proof stage.

Proceedings A levies charges for articles which exceed 20 printed pages. (based upon approximately 540 words or 2 figures per page). Articles exceeding this limit will incur page charges of £150 per page or part page, plus VAT (where applicable).

Under the terms of our licence to publish you may post the author generated postprint (ie. your accepted version not the final typeset version) of your manuscript at any time and this can be made freely available. Postprints can be deposited on a personal or institutional website, or a recognised server/repository. Please note however, that the reporting of postprints is subject to a media embargo, and that the status the manuscript should be made clear. Upon publication of the definitive version on the publisher's site, full details and a link should be added.

You can cite the article in advance of publication using its DOI. The DOI will take the form: 10.1098/rspa.XXXX.YYYY, where XXXX and YYYY are the last 8 digits of your manuscript number (eg. if your manuscript number is RSPA-2017-1234 the DOI would be 10.1098/rspa.2017.1234).

For tips on promoting your accepted paper see our blog post: <https://royalsociety.org/blog/2020/07/promoting-your-latest-paper-and-tracking-your-results/>

On behalf of the Editor of Proceedings A, we look forward to your continued contributions to the Journal.

Sincerely,
Raminder Shergill
proceedingsa@royalsociety.org

Appendix A

Proceedings A RSPA-2021-0210

Bull et al., Modelling of stress transfer in root reinforced soils informed by 4D X-ray computed tomography and digital volume correlation data

Table of revisions

Comment	Response
Referee 1	
This manuscript presents a study of the changes in shear resistance and soil deformations during direct shear due to roots using the no-destructive technique of X-ray CT and digital volume correlation. The authors carried out 3D analysis of local displacements and strains on root-reinforced soils and unrooted. Conventional measurements are also made to complement CT data. The subject of the paper is interesting for the audience of Proceedings A. The paper has interesting results, but before its acceptance for publication, the authors need to clarify some points.	Thank you for the positive comments on the manuscript.
I suggest including some more results in the abstract. The authors only presented a general overview of the results in the abstract, but I believe that some more specific results could be interesting for the readers. Please, insert a last sentence in the abstract highlighting the novelty of the results obtained.	This is a good point. The abstract has been revised to include more detailed findings (e.g. increase in shear zone thickness, the relative maximum reinforcement measured, the root length assumed in the Waldron model versus that measured, and the effect of branching on the soil-root interface shear resistance). It now also includes a statement on the novelty of the results.
The introduction is well written, however, my suggestion is to move the theoretical description of analytical models such as the paragraphs containing equations and figures to a new section, perhaps a theory section? Or simply to incorporate this information in the methodology.	Agreed; the mathematical description of the models is not strictly required in the introduction, and it does possibly break the flow of the narrative. We have chosen to move this information to the methodology, as section 2.5 (on page 7 of the revised manuscript).
The methodology needs some clarification. Please, give more details about the soil preparation in the tubes.	Some further details have been added on page 3, to describe the mixing of the soil, and further information about its compaction into the tubes.
Do the authors believe that only two replications are enough for the CT studies?	We set out to have five replicates of each of the fallow, gorse and willow samples, but had to accept that we did not have enough time resource to run all of these in the XCT scanner, and carry out the associated DVC and other image analyses. As may be expected, there is some variability in the plant-rooted samples because the plants tend to develop and grow slightly differently in each specimen. In each case, the two scanned specimens are sufficient to indicate the broad behaviours that form the novel conclusions of

	the paper (such as the growth of the shear surface around larger roots). The three remaining replicates, which were not XCT-scanned, provide further data on the range of root area ratios, and associated shearing resistances that help to put the XCT-scanned samples in context. A sentence has been added to the paper to note this limitation (page 3).
What was the purpose of the matric potential measurements in the CT?	A tensiometer was pushed into the soil towards the top of the sample prior to the shearing test taking place. It was used to: i) make sure that any (potentially quite large) plant generated suctions were fully removed in the specimen saturation process; ii) that the suctions in each of the samples were about the same, as the suction will influence the effective stress and thus shearing resistance of the soil; and iii) make sure that the suction did not change during the long scanning experiments (it did not, helped by various measures to help prevent any drying from the sample). Some additional comments have been added to the top of page 5 of the revised manuscript to clarify these points.
How to know that the matric potential was the same across the entire sample?	The soil samples were completely saturated (flooded in a water bath) and then allowed to drain under gravity on a bed of wet sand prior to their being sheared in the apparatus. This should have ensured that the samples contained just a small pore water suction throughout, and the tensiometers appear to confirm a very consistent small suction close to the top of the tube between specimens (Table 5; the exception being Willow F, where the stem and leaves of the plant were not removed during the experiment). There may be some small variability of suction in the specimens due to heterogeneity in the soil (section 3.1.1 of the paper), and its disturbance by the plant root system. We did not want to have tensiometers inserted close to the shear plane as this may have interfered with one or both of the shearing process (the tensiometers have a 50 mm long plastic stem that inserts into the sample), and XCT scan (where it is not desirable to have lower density materials in part of the image).
Any influence of the step differences presented in Table 2 in the CT results among treatments?	There were some difficulties getting the first step to be a consistent size, as it was difficult to set up the specimen in the apparatus without any slack in the system. Results have been zeroed (for displacement) at the point at which load is first measured. However, all of the intermediate applied displacement steps are about 3.3 mm, and steps of this size were applied until the box ran out of travel at around 20 mm. As the results are generally comparing an overall trend, or the change in behaviour is often between steps 3 and 7, the effects of the slightly different pause points on the interpretation of the results is small. A comment on the size of the steps has been added on page 5.

Was the voxel resolution obtained enough for the study proposed?	The voxel resolution is a result of a number of XCT scanning equipment technical parameters, but is also fundamentally affected by the size of the soil sample, its density, and its proximity to the X-ray source. To get a smaller voxel resolution, it would have been necessary to compromise on other areas such as the diameter of the sample, and the magnitude of shear displacement that could be applied. This is all a compromise, and is discussed in further detail in our earlier methodology paper (reference [21]). The voxel resolution is sufficient to be able to successfully segment larger roots and carry out DVC analysis that is able to capture the global and some local aspects of soil displacements and strain within the rooted soil, and draw important conclusions from these. It was not sufficient to enable segmentation of the full root systems including finer lateral roots, nor look at very local interactions between the soil and roots. The limitations of the voxel resolution on the ability to segment the full root system is now described in section 2.4 on page 7 of the revised manuscript.
Please, insert a reference for the DaVis LaVision software.	The reference has been inserted.
As the digital volume correlation is one of the techniques utilized in the paper to support the author's findings; I suggest including the description of the methodological steps followed in the paper rather than making reference to another paper.	Several sentences have been added to page 6, to further explain DVC, and to give a brief overview of the studies carried out (and described in more detail in the earlier paper, reference [21]) to understand how to run and optimise the DVC for this set of experiments.
How was determined the two knee points on the tri-linear profile? To me, the procedure presented in Fig. 3 seemed a little bit subjective?	We trialled a number of different methods of calculating the shear band height, including trying to use changes in measured shear strain etc. The method used was selected, as is it gave a locally spatially very consistent measurement of shear band height. Given a wide range of displacement profiles across the sample, we don't think any method will be perfect. The idea is to capture the general size variation across the sample (from 2 mm to 30 mm in height) and the growth trend. The method applied was able to do that. Further comment to this effect has been added to the end of the appropriate section on page 7, and further information on the other methods trialled has been put into a new Supplementary Information for the paper.
In my option, there is an excessive number of figures in the paper, twenty! I suggest to the authors to merge some of them, at least those dealing with similar findings. The same comment is worthy for the tables. Perhaps, some of the figures and tables would included as supplementary material.	The number of figures has been reduced to 16. One figure has been removed (the original Fig. 5) and placed into the Supplementary Information, and three other pairs of like figures have been combined to create the revised Figs. 2, 4 and 15.
What is the explanation for the discontinuities shown in the behavior of the shear stress curves for some of the	The discontinuities are caused by stress relaxation/creep of the root and soil when shearing (applied displacements) is paused so that a XCT scan

treatments presented in Fig. 4? Why some treatments presented this behavior and others do not?	can be carried out. Six shear tests were XCT scanned, and these include the stress relaxation in the plotted shear stress-displacement plots; the remaining samples were sheared continuously outside of the XCT scanner. The original manuscript included a short comment on the stress relaxation in section 3.11, but this has now been extended to make this clearer (see bottom of page 8).
Were the differences in the soil bulk density, matric suction, and water content significant among treatments?	Apart from Willow F, the differences in these measures were small and typical of preparation of geotechnical soil specimens. They are unlikely to be significant compared to variations in the root mass between samples. A comment and reference to this effect has been added to the end of paragraph one on page 9.
Any possible effect of the not resolved roots by CT in the results observed?	There are a large number of smaller roots in some specimens that were not resolved by the XCT, and these will have a significant effect on the shearing of the specimen. A new section, 2.4, has been added to Page 7 to explain more clearly within the manuscript why only some roots could be segmented, and why the macrograph measurements were needed to ensure that all the roots on the shear plane were captured.
If possible, I suggest to the authors merge the results and discussion sections. Some of the results presented were not covered in the discussion. I suggest to the authors to improve them a little bit in the discussion section. Perhaps, it is necessary to work a little bit more on it to make clear the relationships among all the results presented in the study.	The idea of the discussion is to comment on the major findings of the work, and to relate these to the performance of models, limitations in understanding, and larger scales (i.e. the full-scale slope) and engineering practice. The second referee has asked for more comment on some of these, and we have expanded the discussion section; we think it is best left as a separate section for clarity. We have made a series of other improvements to both the methodology and results section as a result of comments made by both referees. These changes will hopefully make the results easier to follow and understand.
Overall evaluation: for the reasons stated above, the manuscript needs additional revision.	
Referee 2	
In this work, the authors evaluate shear resistance from root structures in root reinforced soils by X-ray computed tomography and digital volume correlation. Previous studies investigating the mechanisms of soil-root reinforcement focus on behavior at boundaries due to experimental challenges. This study uses a novel approach developed recently by the authors (ref 21) that uses X-ray CT and DVC to investigate shearing and deformations in	This is a good summary of our work, and thank you for the comments that you have made on the manuscript below.

bulk soil. Three types of specimen were tested: willow, gorse, and fallow. A strong correlation between increased root area ratio in the shear zone and increased shear stress was revealed. Analysis by DVC indicates shows a non-uniform shear zone within the cylinder of soil; instead, shear zone thickness tapers off towards the edges. Roots greater than 1 mm in diameter lead to an observable local increase in shear zone thickness. To better understand the influence of lateral roots in anchoring the root, the authors utilize ABS plastic filament and aluminum discs to simulate non-branching and branching roots. The anchored filament showed a shear stress response curve similar to the rooted specimens suggesting that anchoring plays an important role in soil-root reinforcement. Introduction of variability in shear zone thickness by space and with shear displacement improves predictions of shear stress response by the Waldron model.

The work presents a clear advance by showing how X-ray CT and DVM experiments can be used to refine and improve models of root-reinforced soil shearing. However, important methodological details were published in reference 21 but not summarized here in enough detail, making it difficult to follow the results and understand the data by only reading this manuscript (for example, nowhere in the manuscript I can find the explanation size variation across the sample (from 2 mm to 30 mm in height) and the general growth trend of ϵ_{xz} and ϵ_{xx}). It appears some of the figures were re-used - top panel of Figure 8, left panel of Figure 15 (the examples of CT images) are the same as Figure 13 and 15 in ref. 21. Overall, it would benefit the reader to have some material, such as the variation in fallow tests and root count results sections as well as some figures, moved to supplemental information to streamline the narrative.

Given the relatively high-resolution of the CT scanning, it was puzzling why these data weren't used to greater effect to measure the roots within and without the shear area (including even the root system architecture

Further methodological detail has been added to the manuscript, in response to comments from both reviewers; see pages 3, 5, 6, and 7 of the revised manuscript. The variation in the height of the shear band across the sample in direct shear is a well-established phenomenon; it has been investigated both experimentally and numerically, and others have carried out careful analysis of the stresses that cause the band to form in the way that it does. We comment on this (see the bottom of page 12) with reference to three of the classic papers in this area. We have also made some small edits to the paragraph to hopefully make it clearer. We do have plots of ϵ_{xz} and ϵ_{xx} with increasing applied shear, but don't think that they add anything to the story. The patterns of strains and displacements are clearest at the final displacement step, shown in Fig. 6.

The sections on the root count in the methodology and results have been changed to seek to make the manuscript clearer, including joining the original Figs. 6 and 9 together as a revised Fig. 4, so that they take up less space, and like results are presented together. We have followed the reviewer's suggestion that the section covering the variation in the fallow tests be shortened and the detail and associated figure moved to a Supplementary Information section.

Further information about segmentation of the roots, and the issues associated with it, has been added to a revised Section 2.4 on Page 7. It was found to be possible to visually identify and segment roots greater than 1.5 mm in diameter within the scanned zone of

or other measures of connectivity). Please explain why analysis of the roots (Figure 6) was limited to the microscope experiment, and more details on how the measurements were taken from it. If there are major impediments to using 3D CT data, please note and discuss the potential implications of understanding shearing processes with the entire root system in mind. For example, whether it would be interesting to analyze the relationship between shear stress and other root measurements besides root area ratio.	the sample, but the density and resulting greyscale contrasts were not good enough to be able to identify and segment finer roots. The main difficulty is that the roots have a very similar density to the pore water making it difficult to distinguish between them; this is a common problem in scanning and image analysis of root and soil systems. Comments are made above in reply to Referee 1 as to why the voxel size could not really be any smaller.
Section 3.1.2: The authors only analyzed the x-z slice. Without comparing the 3D dimensions, how can we be confident that the peds in Fallow D are larger than in Fallow P? Page 11, Line 30-34: How can the authors make these conclusions from the results shown here? If these conclusions are from previous publications, the authors should add references.	The commentary on this has been moved to the Supplementary Information, at the suggestion of the referee. A number of vertical thick slices (i.e. averages of several voxels in the out-of-plane direction) were checked (even if only one is shown), and in different orientations. Across the images, the peds were found to be much larger in Fallow D, than in Fallow P. In the commentary on the effects of the size of the peds on the shearing behaviour, we were being speculative, without making this totally clear. We have now edited this part of the paragraph.
Figure 11 and Figure 13. How many points (x, y position) were analyzed for different locations? It looks like each line represent one point on the shear plane. How was the point selected?	These are now Figures 8 and 10. In Figure 8, the displacements are from the DVC, and are taking the difference from last to first scans. In both figures, the points are from locations on or as near as possible to the centre line of the segmented roots, from an x-y position about 5 mm away from segmented roots in the direction of the centre of the tube (marked on plot as 'away from root'), from the centre of the tube (fallow specimens), and from the edge of the DVC field of view (which did not extend to the edge of the specimen; see Figure 4). The text on page 14, and the associated figure captions, have all been edited to make this clear.
The authors observe roots gathered around the edges indicative of a pot-grown system. The deflected lateral roots likely increase reinforcement and are connected to the roots in the center of soil sample. Could the authors please comment on whether deflected roots at the outer boundary influence the measured root-soil interface shear stress?	The pot grown nature of the system will mean that roots will be directed downwards, and this may increase the root area ratio (RAR) within the specimens compared to the field condition. It may also limit the growth of laterals. This would then clearly influence the shearing resistance. A further comment has been made at the top of Page 11 to acknowledge the effect on RAR, and a comment has been made in discussion of the results of the Waldron model (page 25) in relation to the development of the laterals. A further comment to the vertical nature of the roots has been added within the discussion section (page 26).
It appears from Figure 1 and in testing of root analogues that there is an assumption	The pot bound nature of the specimens mean that most of the larger segmented roots were found to be

that roots grow more or less straight down, but they often do not. What implications might this assumption have? An additional figure or diagram describing the analogue experiment would be helpful, and please consider moving these experimental details from the results to the methods section.	sub-vertical in the CT volume; we did also stick to vertical roots with the analogue tests. We agree this may not accurately represent the real field condition. It is possible that analogues could be used to explore the performance of roots that are angled with or against the direction of shearing. A further comment has been added in the discussion to note these points (Page 26). We have chosen not to add a figure to show the analogues, as Referee 1 is concerned about the total number of figures in the manuscript. However, we have edited the section describing the specimens to ensure that their arrangement is clearly described. The details of the analogue specimens have been brought forward and are now described under specimen preparation in Section 2.1 on Page 3. The analogue specimens have also been listed in Tables 1 and 5, as they were not previously included.
In the anchored filament measurements, resistance introduced by anchoring is evenly distributed above and below the shear plane. However, root systems are likely to have would have different lateral branching above and below the shear plane. How would asymmetry in anchoring/branching affect root-soil interface shear stress?	Yes, this is a good point. Of course, if friction also plays a role in the root resistance then the shear stress at the root-soil interface might be expected to increase with depth, with the normal effective stress in the soil. Further comment to both points has been added to the discussion section on Page 26.
The authors acknowledge their uncertainty on whether variability in shear zone thickness would be observed in larger systems. It would benefit the reader to have this uncertainty and other differences from larger and/or sloped systems further discussed in the discussion. In this study, shear strain is applied planar to the ground. In a sloped system, plants have asymmetric root architecture to accommodate differences in the distribution of mechanical forces into the soil. How would such an architecture affect shear stress response?	We agree that that it would be helpful to comment also on the field scale differences in the root system. Further comment has been added to the end of the discussion section to note these points (Page 26).
Section 2.4, Page 8, Line 38: What are the several different calculation approaches?	There were three other calculation approaches. As both referees have asked about these, they are now described in the Supplementary Information. We did not wish to put the full details in the paper, as the approaches take quite a number of words and two figures to explain comprehensively.
In figure 12, please include the position of roots >1 mm in diameter as shown in figure 9 and 10.	These have been added to the figure.